# Genetic diversity, determinants, and dissemination of *Burkholderia pseudomallei* lineages implicated in melioidosis in Northeast Thailand

Rathanin Seng[1], Chalita Chomkatekaew [2,3], Sarunporn Tandhavanant [1], Natnaree Saiprom[1], Rungnapa Phunpang[1], Janjira Thaipadungpanit[4], Elizabeth M. Batty [2,5], Nicholas P. J. Day[2,5], Wasun Chantratita[6], T. Eoin West [7,8], Nicholas R. Thomson [9,10], Julian Parkhill [3], Claire Chewapreecha [2,4,9,11,12] ✉ & Narisara Chantratita [1,2,12] ✉

Melioidosis is an often-fatal neglected tropical disease caused by an environmental bacterium *Burkholderia pseudomallei*. However, our understanding of the disease-causing bacterial lineages, their dissemination, and adaptive mechanisms remains limited. To address this, we conduct a comprehensive genomic analysis of 1,391 *B. pseudomallei* isolates collected from nine hospitals in northeast Thailand between 2015 and 2018, and contemporaneous isolates from neighbouring countries, representing the most densely sampled collection to date. Our study identifies three dominant lineages, each with unique gene sets potentially enhancing bacterial fitness in the environment. We find that recombination drives lineage-specific gene flow. Transcriptome analyses of representative clinical isolates from each dominant lineage reveal increased expression of lineage-specific genes under environmental conditions in two out of three lineages. This underscores the potential importance of environmental persistence for these dominant lineages. The study also highlights the influence of environmental factors such as terrain slope, altitude, and river direction on the geographical dispersal of *B. pseudomallei*. Collectively, our findings suggest that environmental persistence may play a role in facilitating the spread of *B. pseudomallei*, and as a prerequisite for exposure and infection, thereby providing useful insights for informing melioidosis prevention and control strategies.

Melioidosis, a severe infectious disease, affects an estimated 165,000 cases globally each year, of which 89,000 are fatal[1]. The disease is caused by *Burkholderia pseudomallei*, a Gram-negative bacillus found in soil and contaminated water across tropical and sub-tropical regions. Historically, limited access to microbiology laboratories for culture-confirmed diagnosis led to underreporting, particularly in lower- and middle-income countries[2]. However, improved infrastructure and awareness have led to increases in reported cases across South Asia, Southeast Asia, East Asia[3–8] and Australia[9,10] In Southeast Asia, the disease incidence is often linked to agriculture practice,

---

particularly during the rainy seasons when rice paddy fields are flooded for planting[11]. The flooded terrain enables the bacterium in the soil to surface, potentially exposing farmers to *B. pseudomallei* and subsequently leading to melioidosis. Additionally, many cases of melioidosis have been associated with severe weather events[12–14]. While climate likely influences human encounters with *B. pseudomallei*, further investigation is needed to fully understand the mechanisms linking environmental factors to melioidosis epidemiology.

Understanding the population structure, dissemination and adaptation of *B. pseudomallei* in these climatically-challenged endemic regions requires a large-scale, geographically and chronologically densely-sampled, genetic dataset. Previous studies, albeit limited in sample size, have demonstrated that *B. pseudomallei* dissemination is driven by both anthropogenic and environmental factors[15–18]. Streams[19,20], monsoons, typhoons and cyclones[13,14,21,22] were identified as significant contributors to bacterial dissemination, highlighting the importance of bacterial persistence across a range of environmental conditions. *B. pseudomallei* exhibits remarkable survival capabilities across diverse environments, spanning from wet to dry, nutrient-depleted soil[23–28] thereby enabling the bacterium to thrive in various ecological niches. Previous studies have noted the temporal and geographical co-existence of multiple *B. pseudomallei* lineages[15,16,29].

However, little is known about their distinct genetic content and adaptive strategies. Identification of lineage-specific genes associated with bacterial persistence and disease escalation will be essential to develop disease control strategies.

In this study, we conducted a population genomics analysis using combined *B. pseudomallei* isolates from melioidosis patients across nine provinces in the northeast Thailand including Buriram, Khon Kaen, Mahasarakham, Mukdahan, Nakhon Phanom, Roi Et, Sisaket, Surin and Udon Thani[8]; totaling 1265 isolates collected from July 2015 to December 2018. Additionally, we incorporated contemporary environmental and clinical collections from Thailand and neighbouring countries[29–34], consisting of 15 clinical isolates and 111 environmental isolates (Fig. 1a, Supplementary data 1). Our comprehensive analysis, including a total of 1391 isolates, revealed the population structure, dissemination patterns, and genetic diversity of this bacterium. We identified genetic determinants associated with dominant lineages and investigated their biological functions and expression conditions in three representative isolates, each representing a dominant lineage. This provides insights into the strategies employed by *B. pseudomallei* lineages for successful persistence in the environment, ultimately leading to human exposure and infection.

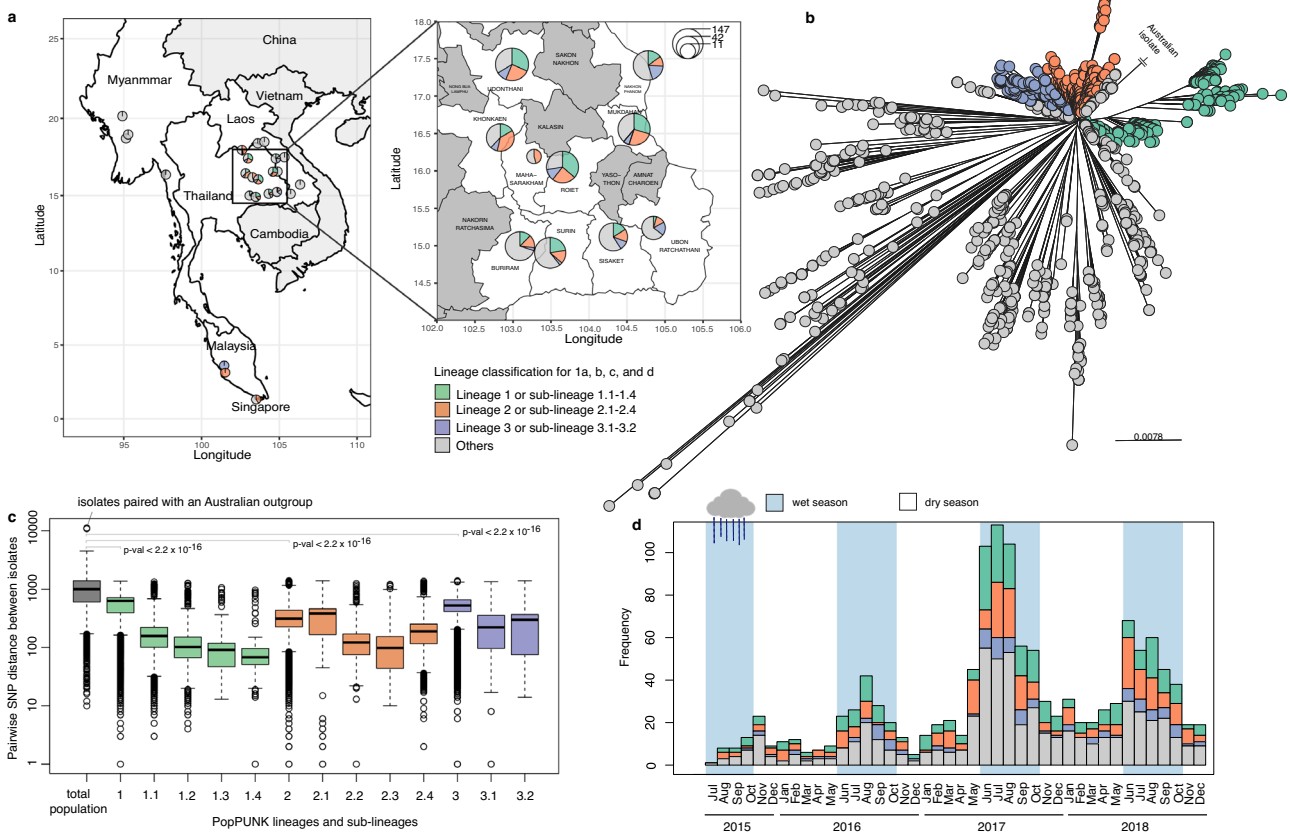

**Fig. 1 | Distribution of *B. pseudomallei* genomes used in this study.**
**a** Geographical representation of the countries and provinces sampled for the 1391 *B. pseudomallei* genomes used in this study. Pie-chart summarises the proportion of dominant lineage 1, 2, and 3 presented at each location with the chart size proportional to the number of the samples collected **b** An unrooted phylogenetic tree colour-coded by dominant lineages **c** Boxplots summarising the pairwise core genome SNP distances among isolates in this study are shown on a logarithmic scale. The distributions are depicted for the entire population, each dominant lineage and its sub-lineages. Each boxplot presents the minimum, first quartile, median, third quartile, and maximum data points. The number of pairs in each boxplot ranges from 105 to 968136, which corresponds to the smallest group

combination of C (15, 2) in sub-lineage 1.4, and the pair combination of the entire population and an Australian outgroup C (1392, 2). The difference in pairwise SNP distributions between isolates clustered across different lineages within the entire population and those clustered within each defined lineage was calculated using a two-sided Mann–Whitney U test. **d** Histogram depicting the distribution of clinical *B. pseudomallei* isolates from the northeast Thailand cohort throughout 2015–2018 sampling period. For (**a**–**d**), isolates are colour-coded: lineage 1 and sub-lineage 1.1–1.4 in green, lineage 2 and sub-lineage 2.1–2.4 in red, lineage 3 and sub-lineage 3.1–3.2 in purple, and other lineages in grey. In (**d**), the shaded blue and white areas represent the wet and dry seasons in northeast Thailand, respectively.

## Results

### Population structure analysis identified successful *B. pseudomallei* lineages and the intermixing of clinical and environmental isolates

To define the population structure of clinical and environmental *B. pseudomallei* in a hyperendemic area of northeast Thailand and neighbouring regions (*n* = 1391), we performed four independent approaches. PopPUNK[35] analysis was performed on genome assemblies (Supplementary data 2). Additionally, we constructed three maximum-likelihood (ML) phylogenies[36], each based on different sets of single nucleotide polymorphisms (SNPs): core genomes (*n* = 77,156 SNPs), core gene multilocus typing[37] (cgMLST, *n* = 46,945 SNPs), and seven-gene multilocus typing genes[38] (MLST, *n* = 31 SNPs). These approaches facilitated the grouping of isolates with close genetic similarity into distinct lineages. Notably, both PopPUNK analysis and core genome SNPs phylogeny yielded consistent results (Supplementary Fig. 1) clustering the population into three dominant lineages (Fig. 1b). The average pairwise core SNP distance within each dominant lineage was 549, 351, and 517 SNPs for lineage 1, 2, and 3, respectively, in contrast to the average pairwise core SNP distance of 1087 SNPs within the total population (Fig. 1c). This lower pairwise core SNP distance across lineages confirmed the genetic relatedness as defined by PopPUNK and core genome SNP phylogenetic analysis. While cgMLST displayed conservation for two out of three dominant lineages, MLST exhibited inconsistencies across dominant lineages with lower phylogenetic resolution and poorer bootstrap support compared to other methodologies (Supplementary Fig. 1). Consequently, we relied on the population delineated by PopPUNK and core genome SNP phylogeny for subsequent investigations.

The three predominant lineages (denoted as lineage 1 to 3) comprised 312, 297, and 125 isolates, respectively. They accounted for 52.8% of the studied population and persisted throughout the sampling period. Interestingly, each lineage peaked during the wet season, correlating agricultural practices at the onset of rainfall with increased environmental exposure and subsequent melioidosis infections (Fig. 1d). Despite the small sample size of environmental isolates, we observed a clustering of these isolates with clinical isolates within each dominant lineage, indicating their core genetic similarities and shared origin. The ratio of environmental to clinical isolates varied across lineages (Chi-square test with Monte Carlo resampling *p*-value $5.00 \times 10^{-4}$, Supplementary Fig. 2). Due to the substantially lower number of environmental isolates used and incomplete geographical distribution matching between clinical and environmental isolates, caution is warranted in interpreting these results. Nevertheless, our findings highlight a mixing of environmental and clinical samples, suggesting that clinical isolates could serve as a surrogate for tracking the dissemination of an environmental bacterium, especially in the absence of equally comprehensive environmental samples.

### Genetic evidence identifies patterns of *B. pseudomallei* dissemination in Northeast Thailand

We examined the dissemination patterns of *B. pseudomallei* in northeast Thailand at the provincial level. Except for Mahasarakham where samples were limited, all three dominant lineages were present across the rest of the eight studied provinces (Fig. 1a). This prompted us to focus the analysis on these dominant lineages to identify consistent geographical distributions underlying their spread in the region. We generated lineage-specific phylogenies to improve genetic resolution for transmission analysis. Additionally, we reconstructed ancestral histories of provincial origins, quantified the number of interprovincial transmissions to examine transmission patterns, and estimated the time of the most recent common ancestor of each dominant lineage and its sub-lineages (Supplementary Figs. 3–6, see "Methods"). This allowed us to link the emergence of these lineages to historical events that might have affected their transmission dynamics. While

transmission signals could reflect distinct dissemination patterns in each dominant lineage or its sub-lineages, we also considered the possibility of shared factors leading to a uniform geographical distribution. Notably, we observed consistent dissemination patterns in 14 out of 28 provincial pairs across the three dominant lineages (Fig. 2a, Supplementary Fig. 6). Eight out of these 14 provincial pairs showed potential correlation with the slope of terrain altitude between provinces or the natural flow of rivers in the region (Fig. 2b). These pairs included "Udon Thani-to-Khon Kaen", "Udon Thani-to-Buriram",, "Udon Thani-to-Surin", "Udon Thani-to-Mukdahan", "Khon Kaen-to-Buriram", "Nakhon Phanom-to-Mukdahan", "Roi Et-to-Surin" and "Surin-to-Sisaket". Northeast Thailand is described as a saucer-shaped plateau, with elevations ranging from over 200 meters above sea level in the northwestern corner (parts of Udon-Thani, Khon-Kaen and Buriram) to less than 140 meters in the southeast (parts of Mukdahan and Sisaket), and gradually descending toward the Mekong River in the east[39,40]. The Mun River originates from elevated hills in central Thailand, streaming eastward through Buriram, Surin, Sisaket before merging with the Mekong River. Similarly, the Chi River, a tributary to the Mun, also originates from the central Thailand mountains. The Chi flows eastward through Khon Kaen, Mahasarakham, Roi Et and converges with the Mun in Sisaket[40]. These geographical features may, at least in part, explain the observed dissemination patterns.

Additionally, another set of three out of 14 conserved patterns coincided with the wind direction of the northeastern monsoon during the dry season ("Nakhon Phanom-to-Buriram", "Mukdahan-to-Buriram", and "Mukdahan-to-Surin"). Thailand experiences two predominant monsoon seasons: the southwest monsoon from approximately May to October and the northeast monsoon from approximately November to April. The southwest monsoon brings heavy rainfall from the southwest to the northeast, often marking the start of the agriculture season across Thailand (Fig. 1d)[41,42]. Conversely, the northeast monsoon brings dry winds from the northeast to the southwest. Our observation implies that dry winds may have the potential to transport aerosolised soil contaminated with *B. pseudomallei* southwestward. Despite that previous air sampling during Thailand's wet season did not detect *B. pseudomallei*, exploring the impact of dry winds during the northeastern monsoon is essential. This consideration becomes even more pertinent given reports of the long-range transport of particles and small organic matters, such as PM2.5 and PM10, via the northeast monsoon elsewhere in Southeast Asia[43,44]. Furthermore, we successfully estimated the time of most recent of ancestry for a sub-lineage 1.3, a descendant of lineage 1. Our finding revealed that this sub-lineage emerged around 2011 (95% HPD of 2000–2014, Supplementary Fig. 3c). The age of this sub-lineage implies that older lineages, such as its parental lineage 1 likely experienced multiple monsoon seasons, which possibly resulted in the observed patterns. Although our observation suggested terrain slope, inland rivers, canal systems[45] and regular monsoons[41,42] as the potential factors that may contribute to *B. pseudomallei* dissemination in northeast Thailand, the result should be interpreted with caution since anthropogenic activities such as human migration between the provinces may also contribute to the observed pattern. However, without access to comprehensive human movement data, this aspect remains challenging to investigate.

### Genetic markers present in successful *B. pseudomallei* lineages

The co-existence of multiple *B. pseudomallei* lineages within the same geographical areas and timeframe implies the presence of diverse adaptive strategies that enable them to thrive in a shared ecological niche. While some smaller lineages may be sporadically detected, the persistence of the three dominant lineages throughout the sampling period supports their fitness and successful adaptive strategies in this niche. We next sought to identify genes that were present in isolates that form each dominant lineage, or its sub-lineage; but absent in

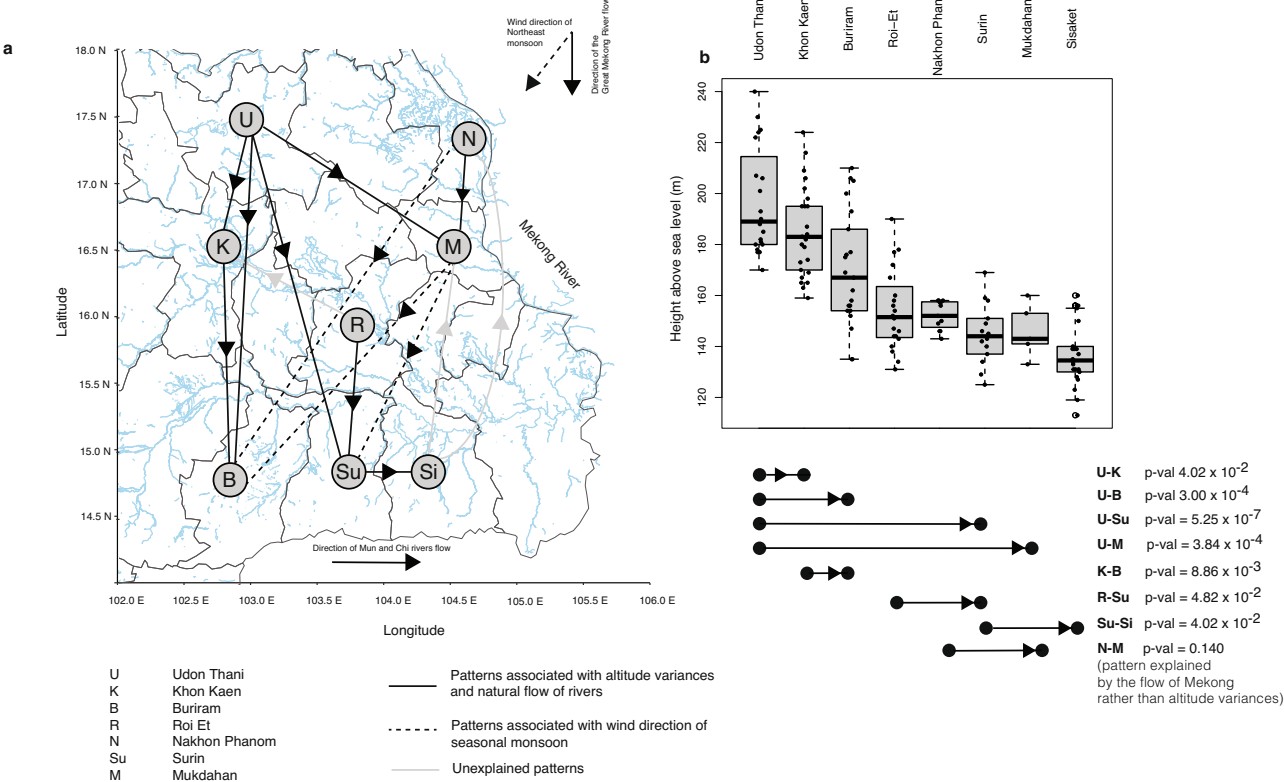

**Fig. 2 | Dissemination patterns in Northeast Thailand. a** Province-to-province transmission patterns potentially influenced by northeast Thailand geographical landscape. Nodes present provinces, denoted by abbreviation and ordered by average altitude: U Udon Thani, K Khon Kaen, B Buriram, R Roi Et, N Nakhon Phanom, Su Surin, M Mukdahan, and Si Sisaket. Rivers are depicted in blue with major rivers including the Great Mekong River, the Chi River, and the Mun River and their flow direction annotated. Only transmission patterns consistently observed in at least two out of three dominant lineages are shown. **b** Distribution of the terrain height for each province in meters above sea level. Each dot represents the height of the district where patients lived, indicating the closest source of bacterial acquisition. Each boxplot presents the minimum, first quartile, median, third quartile, and maximum data points. Sample size for each province are as follows: Udon Thani ($n = 20$), Khon Kaen ($n = 25$), Buriram ($n = 21$), Roi Et ($n = 20$), Nakhon Phanom ($n = 11$), Surin ($n = 14$), Mukdahan ($n = 5$) and Sisaket ($n = 22$). The northwest provinces have higher altitudes, which gradually decline towards the southeast. This province-to-province pattern is confirmed by a one-sided Mann–Whitney $U$ test, with $p$-values annotated on the figure. For (**a**) and (**b**) solid arrows illustrating transmission directionality possibly explained by altitude differences. Dotted arrows represent transmission directionality potentially influenced by northeast monsoon winds. Grey arrows signify patterns with unclear explanation.

non-dominant lineages (see "Methods", Supplementary Fig. 7). Out of a total of 15,237 genes in the pan-genome outlined from this population (see "Methods"), 5577 genes were conserved across the entire population while 9660 genes were variably present (accessory genes). Dominant lineage-specific genes were defined as accessory genes present in ≥ 95% of isolates within any of the dominant lineages or their sub-lineages, but present in ≤ 15% of isolates outside these lineages. Among these, 247 genes were identified as lineage-specific with their specificity to each dominant lineage and sub-lineage tabulated in Supplementary Data 3. The majority of dominant lineage-specific genes were poorly characterised and annotated as hypothetical proteins (Fig. 3). To gain insights into the potential functions, we annotated them using Gene Ontology (GO terms)[46] which classify them by Biological process, Molecular function, and Cellular component (Fig. 3b, see "Methods"). Of the 247 dominant lineage genes, GO terms could be assigned to 42 genes for Biological Process, 68 for Molecular Function, and 12 for Cellular component. For genes that could be assigned GO terms, functions involved in "DNA integration", "DNA recombination", and "DNA methylation" might indicate their potential roles in horizontal gene acquisition and protection against incoming foreign DNA through site-specific DNA methylation. Furthermore, GO terms associated with "DNA binding" and "Regulation of DNA-templated transcription" may suggest lineage-specific regulation of the expression of these genes.

## Lineage-specific genes were selectively expressed

To delve deeper into the functionality of dominant lineage-specific genes, we explored their expression patterns across both environmental and infection conditions. We selected representative strains - "K96243" (lineage 1 – sub-lineage 1.1), "UKMD286" (lineage 2 – sub-lineage 2.1), and "UKMH10" (lineage 3 – sub-lineage 3.2) - all are clinical isolates with pre-existing gene expression profiles under infection and environmental conditions[47–49]. For environmental conditions, K96243 was exposed to water[47], while UKMD286 and UKMH10 were cultivated in a soil extract medium[48,49] to mimic *B. pseudomallei* in the environment. For infection conditions, K96243 and UKMD286 were used in murine challenges[47,48] and isolated from mice organs, while UKMH10 was subjected to human plasma[49] to simulate host infection. This approach facilitated the comparison of differentially expressed lineage-specific genes between environmental and infection conditions. Although each representative strain carried a complete set of lineage-specific genes for their respective sub-lineages (K96243 with 47 genes, UKMD286 with 27 genes, and UKMH10 with 14 genes), their collective representation accounted for 69 out of 247 total lineage-specific genes (27.9%). This limitation was due to observed genetic diversity within the dominant lineage, which should be acknowledged. Interestingly, 11 out of 47 lineage-specific genes in K96243 and 6 out of 27 lineage-specific genes in UKMD286 were up-regulated in the environmental conditions (Figs. 4a–c, Supplementary Data 4). However, none of the lineage-

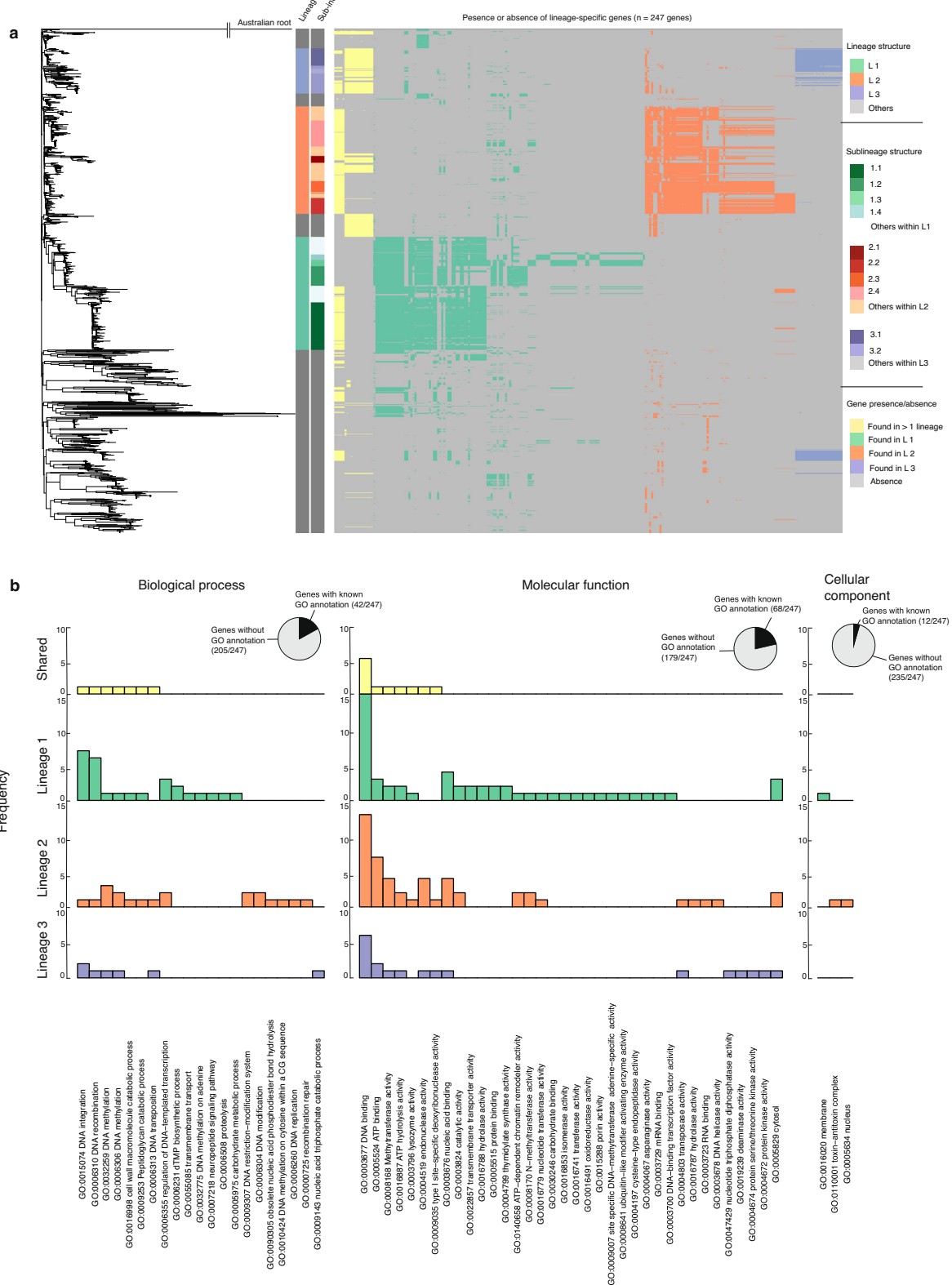

**Fig. 3 | Dominant lineage-specific genes and their gene ontology (GO terms).**
**a** The heatmap represents lineage-specific genes (right) detected in each isolate, aligned with the phylogeny (left). Lineage-specific genes shared across multiple dominant lineages are highlighted in yellow. Lineage-specific genes from lineage 1, 2, 3 are coloured in green, red, and purple, respectively. Additionally, the colour stripes provide information on the lineage and sub-lineage membership (**b**) Bar plots displays the frequency of GO annotations of lineage-specific genes in each dominant lineage categorised by biological process, molecular function, and cellular compartment. The pie-charts summarise the proportion of lineage-specific genes with assigned GO terms (black).

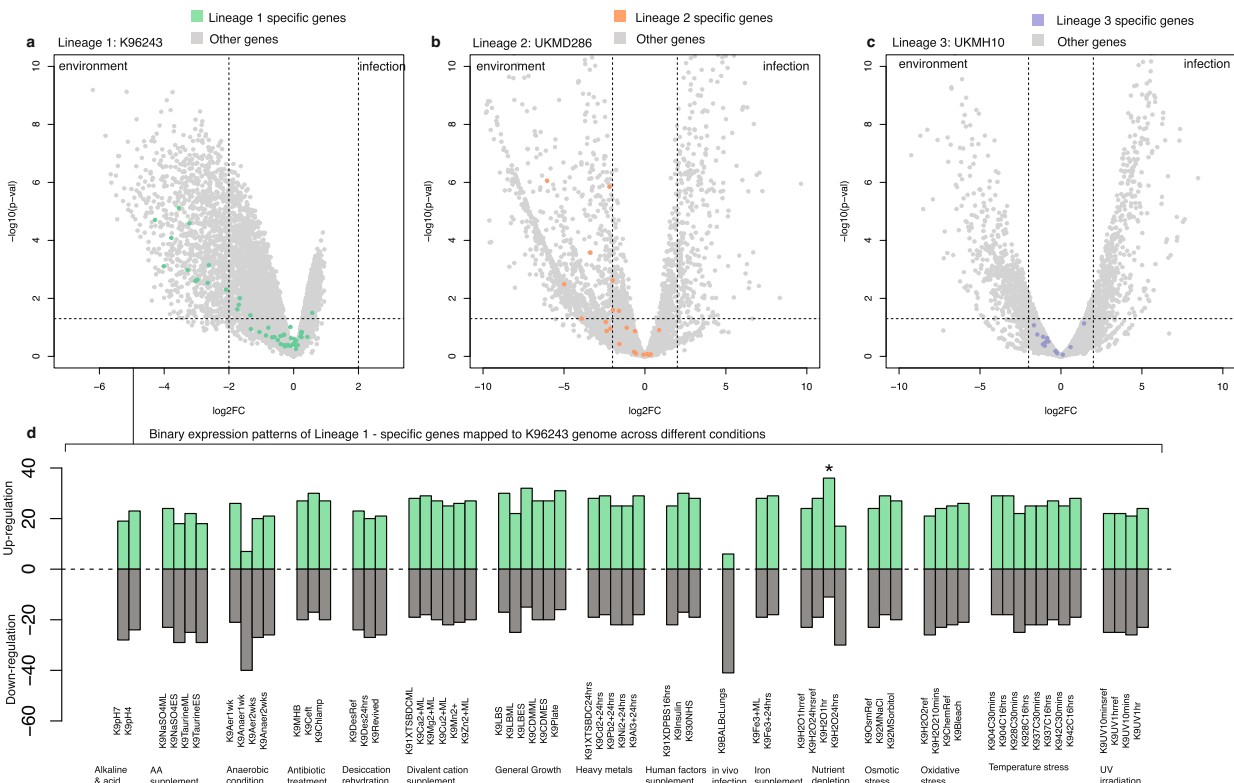

**Fig. 4 | Transcriptome analysis of representative strains: K96243 (lineage 1), UKMD286 (lineage 2) and UKMH10 (lineage 3). a–c** Volcano plots demonstrate differential gene expression (DGE) between environmental and infection conditions. **a** Based on microarray data, RNA probes with significant changes were identified using two-sided $t$-test with multiple test corrections via limma. **b**, **c** Based on RNAseq data, significant changes in RNAs were identified using two-sided Wald test with multiple testing corrections performed by DESeq2. Vertical dotted lines represent the statistical cut-off at an absolute value of $\log_2$ fold change > 2, while horizontal dotted lines display the statistical cut-off at the adjusted $p$-value of 0.05 on a negative $\log_{10}$ scale. **d** Binary expression profile of lineage-1-specific genes across different conditions. Tested conditions are as follows: Alkaline & acid: alkaline solution (K9pH7) and acidic solution (K9pH4); Amino acid supplement: mid-logarithmic phase in free sulfur source (K9NaSO4ML), early stationary phase in free sulfur source (K9NaSO4ES), mid-logarithmic phase in amino acid sulfur source (K9TaurineML), and early stationary phase in amino acid sulfur source (K9TaurineES); Anaerobic condition: 1-week aerobic culture (K9Aer1 wk), 1-week anaerobic culture (K9Anaer1 wk), 2-week aerobic culture (K9Aer2 wks), and 2-week anaerobic culture (K9Anaer2 wks); Antibiotic treatment: medium without antibiotics (K9MHB), bacteriocidal antibiotic (K9Ceft), and bacteriostatic antibiotic (K9Chlamp); Desiccation and rehydration: hydrated bacterial pellet (K9DesRef), desiccated bacterial pellet (K9Des24 hrs), and rehydrated bacterial pellet (K9Revived); Divalent cation supplement: chelated medium (K91XTSBDCML), $Ca^{2+}$ exposure (K9Ca²⁺ML), $Mg^{2+}$ exposure (K9Mg²⁺ML), $Cu^{2+}$ exposure (K9Cu²⁺ML), $Mn^{2+}$ exposure (K9Mn²⁺ML), and $Zn^{2+}$ exposure (K9Zn²⁺ML); General Growth: stationary phase in rich media (K9LBS), mid-logarithmic phase in rich media (K9LBML), early stationary phase in rich media (K9LBES), mid-logarithmic phase in minimal media

(K9CDMML), early stationary phase in minimal media (K9CDMES), and bacterial lawn (K9Plate); Heavy metal: chelated medium (K91XTSBDC24 hrs), $Cd^{2+}$ exposure (K9Cd²⁺24 hrs), $Pb^{2+}$ exposure (K9Pb²⁺24 hrs), $Ni^{2+}$ exposure (K9Ni²⁺24 hrs), and $Al^{3+}$ exposure (K9Al³⁺24 hrs); Human factors supplement: saline (K91XDPBS16 hrs), insulin (K9Insulin), and normal human serum (K930NHS); In vivo infection: murine infection (K9BALBcLungs); Iron supplement: mid-logarithmic phase with $Fe^{3+}$ exposure (K9Fe³⁺ML), and stationary phase with $Fe^{3+}$ exposure (K9Fe3+24 hrs); Nutrient depletion: early exposure to usual nutrients (K9H201href), long exposure to usual nutrients (K9H2024hrsref), initial response to nutrient deprivation (K9H201 hr), and long response to nutrient deprivation (K9H2024 hrs); Osmotic stress: normal osmolarity (K9OsmRef), high salinity (K92MnaCl), and high osmolarity (K92Msorbitol); Oxidative stress: normal growth without hydrogen peroxide (K9H2O2ref), hydrogen peroxide (K9H2O210 mins), normal growth without sodium hypochloride (K9ChemRef), and sodium hypochloride (K9Bleach); Temperature stress: initial response to cold stress (K904C30 mins), overnight incubation under cold stress (K904C16 hrs), initial response to ambient temperature (K928C30 mins), overnight incubation under ambient temperature (K928C16 hrs), initial response to body temperature (K937C30 mins), overnight incubation under body temperature (K937C16 hrs), initial response to heat stress (K942C30 mins), and overnight incubation under heat stress (K942C16 hrs); UV irradiation: 10-min under fluorescent light (K9UV10 minsref), 1-hour under fluorescent light (K9UV1hrref), 10-min under UV irradiation (K9UV10 mins), and 1-hour under UV irradiation (K9UV1 hr). A star denotes significant differences in the gene expression profile of lineage-1-specific genes compared the remaining genes of strain K96243 (adjust $p$-value = $5.72 \times 10^{-3}$, one-sided Fisher's exact test with Benjamini-Hochberg correction for multiple testing).

specific genes showed up-regulation during the infection conditions. The remaining lineage-specific genes did not exhibit preferential expression in either environmental or infection conditions.

The elevated expression level of lineage-specific genes in the environmental condition was unexpected considering that all representative strains were clinical isolates. Our analysis of a broader spectrum of environmental conditions (Fig. 4d) for K96243[47], a lineage 1 representative strain, revealed that lineage-1-specific genes exhibited higher expression levels under nutrient deprivation compared to other genes in K96243 genome (Benjamini Hochberg adjusted $p$-value = $5.72 \times 10^{-3}$). This finding suggests that the ability to survive in

nutrient-depleted soil, which is not uncommon in melioidosis endemic areas[27,28,50] before being acquired by a human host and subsequently causing the disease, maybe one of the adaptive strategies of lineage 1. However, caution is needed in interpreting this finding due to the limited number of characterised lineage-specific genes and the use of a representative strain, which may not fully capture the genetic diversity within the lineage.

### Example of lineage-specific genes
The majority of lineage-specific genes were located within genomic islands (GI)[30,51,52]. These regions are characterised by anomalies in %

**Table 1 | Recombination in dominant lineages by coding sequences (CDS)**

| Lineage | Percent of CDS impacted by recombination at least once (recombinant CDS/total CDSs in the reference genome) | Percent of dominant lineage-specific genes underwent recombination at least once (recombinant gene/lineage-specific genes identified in the reference genome) | Average r/m (number of SNPs introduced by recombination/ SNPs introduced by substitutions) | | |
|---|---|---|---|---|---|
| | | | Internal nodes (95% CI) | Terminal nodes (95% CI) | Average (95% CI) |
| 1 | 99.5% (5981/6010) | 100% (47/47) | 2.8 (2.3–3.3) | 4.6 (4.0–5.3) | 3.7 (3.3–4.1) |
| 2 | 99.9% (5963/5971) | 100% (65/65) | 5.0 (3.9–6.0) | 4.3 (3.6–4.9) | 4.6 (4.0–5.2) |
| 3 | 96.6% (5338/5523) | 100% (20/20) | 1.4 (1.1–1.8) | 2.9 (2.2–3.7) | 2.2 (1.8–2.6) |

G + C content or dinucleotide frequency signatures, or the presence of genes associated with mobile genetic elements such as insertion sequence (IS) elements and bacteriophages. Notably, we observed that a cluster of genes specific to lineage 1 (*BPSS2060* to *BPSS2072*) formed a mosaic structure within a putative metabolic island known as GI 16[30]. Although several variations of GI 16 have been reported (GI16, GI16.1, GI16.2, GI16a, GI16b, and GI16b.1)[51], it typically spans 60 kb (*BPSS2051* to *BPSS2090*) and carries several known virulence determinants and genes that enhance metabolic versatility. While certain virulence factors, such as the filamentous haemagglutinin (*BPSS2053*) required for host cell adhesion and its processing protein were conserved across multiple lineages observed in our study, genes encoding functions that potentially expand the metabolic repertoire were specific to dominant lineages. For example, the mosaic structure of GI 16 (*BPSS2060* to *BPSS2072*), specific to lineage 1 (Supplementary data 3), contains genes involved in alternative nutrient catabolism and anabolism (*BPSS2060*, *BPSS2065*, *BPSS2067*, *BPSS2068*, and *BPSS2072*), transcriptional regulation (*BPSS2061*), and substrate transport (*BPSS2064*, *BPSS2071*). Out of 11 lineage-specific genes located in the mosaic structure of GI 16, eight were found to be upregulated during the early phase of nutrient starvation while remaining silent during infections. This finding reflects the functional division of GI16 where its lineage-specific mosaic structure contains genes that contribute to metabolic versatility, while its core structure encodes virulent determinants associated with disease implications. It is important to note that the structure of GI 16 may vary across different regions due to the plasticity of genomic islands and changes in selection pressures.

## Roles of homologous recombination

Homologous recombination has been shown to play a significant role in facilitating the gain and loss of genes, and the generation of mosaic structures within the GI of *B. pseudomallei*[53]. To better understand its association with lineage-specific genes, we identified recombination events and quantified the rates of recombination in the dominant lineages (Supplementary Fig. 8). The ratio of polymorphisms introduced through recombination compared to those introduced by mutation (*r/m*) was 3.7, 4.6 and 2.2 for lineages 1, 2, and 3 respectively (Table 1). A very high proportion of genes underwent recombination at least once: 99.5% of genes in lineage 1, 99.9% in lineage 2, and 96.6% in lineage 3. Furthermore, every lineage-specific gene within each dominant lineage underwent recombination (Table 1). The bacterial restriction modification (RM) systems prevent the invasion of foreign DNA and restrict gene flow between *B. pseudomallei* lineages[31]. Notably, components of this system including a type I restriction system and modification methylase were among dominant lineage-specific genes (*BPSL0947-BPSL0948* in lineage 1, and their homologues in lineage 2 and 3). They may act as a barrier for homologous recombination and potentially modulate lineage-specific genetic diversity. This highlights the intricate interplay between recombination, lineage-specific genes, and the RM system in shaping the genetic landscape of *B. pseudomallei* in northeast Thailand.

## Discussion

Our analysis of *B. pseudomallei* population genomics enhances our understanding of the evolution and adaptive strategies employed by the dominant lineages in the melioidosis hyperendemic region of northeast Thailand and neighbouring countries. Through an unprecedentedly dense sampling effort between 2015 and 2018, we were able to determine the co-existence of three dominant lineages, characterise their dissemination patterns, and identify lineage-specific genes possibly contributing to their success during the studied period. By analysing transcriptome data from representative strains of each dominant lineage, we gained insights into some of their adaptive strategies, highlighting bacterial persistence in the environment as crucial for subsequent host acquisition and infection, among other strategies. Nevertheless, our study has a few limitations.

While a complete transmission landscape of an environmental pathogen would ideally incorporate both environmental and clinical isolates, our study primarily utilised clinical isolates. This choice was necessitated by the limited environmental surveillance and the scarcity of environmental isolates in Southeast Asia. Although this approach is not ideal, it represents the best available scenario. Importantly, all clinical isolates were most likely acquired from environmental exposures[54], and human-to-human transmission is known to be very rare for melioidosis[55]. Furthermore, the co-occurrence of clinical and environmental isolates within the same lineage suggests that our findings based on clinical isolates may have broader implications for environmental isolates. Nevertheless, integrating environmental studies is needed to enhance our understanding of *B. pseudomallei* persistence in the environment and its transmission. Therefore, continued surveillance efforts including both clinical and environmental samples will be essential. Additionally, the lineage classification report in our study is subjected to potential alterations over time with the introduction of new data. As *B. pseudomallei* continuously adapts to environmental pressures, the composition of lineages 1, 2, and 3, along with their respective lineage-specific genes may shift. New lineages with superior fitness or selective advantages, of different adaptive strategies could potentially outcompete the existing dominant lineages. Consequently, this may lead to emergence of new lineage classifications in the future. This ongoing monitoring will be pivotal in identifying alterations within the bacterial population, uncovering new adaptive strategies, and evaluating their impact on disease dynamics over time.

The up-regulation of dominant lineage-specific genes in environmental conditions, coupled with the increased expression of lineage-1-specific genes during nutrient deprivation, suggests that bacterial persistence in the environment may be an adaptive strategy for certain *B. pseudomallei* lineages. However, it is essential to acknowledge the genetic diversity within each dominant lineage. The representative strains in our study carried only a subset of the lineage-specific genes. As a result, lineage-specific genes absent in these strains, often annotated as hypothetical proteins, remained unexplored. To gain a more comprehensive understanding, we need a fuller spectrum of

conditions reflecting different transcriptome profiles mirroring seasonal fluctuations. Similar efforts have been successful in elucidating bacterial transitions through host cells during infection[56], resulting in a comprehensive list of genes and their functional during different infection stages. Therefore, a broader range of experimental conditions is needed to uncover additional adaptive strategies overlooked in this study and pave the way for future exploration.

Despite these limitations, our dataset and analysis represent one of the most comprehensive efforts to date. Our findings suggest that environmental persistence may be one of the adaptive strategies for dominant lineages 1 and 2, with lineage-1-specific genes potentially mediating bacterial survival under nutrient depletion. It remains uncertain whether lineage 3 adopts a similar strategy. Nevertheless, our results align with previous soil sampling studies, which consistently observed a higher prevalence of *B. pseudomallei* in nutrient-depleted compared to nutrient-rich soil[28,50]. Additionally, a molecular evolutionary study also supports the species' long-term adaptation to survive nutrient scarcity[27]. The northeast region of Thailand, where our samples were primarily collected, has inherently low-fertility soil. This is exacerbated by intensified agriculture, monoculture, excessive synthetic fertiliser use, and poor land management, resulting in depleted soil nutrients and organic matter[57]. This may present a challenging environment for *B. pseudomallei* to thrive, thereby potentially selecting for successful lineages with persistent traits as observed in our study.

Our analyses also highlight various factors such as differences in terrain altitude[58,59], river flow dynamics[40], and the northeast monsoon[41] as the drivers that may shape the dissemination patterns of *B. pseudomallei*. These drivers of dissemination may be influenced by both natural and human activities[60]. For instance, strong winds can carry dried soil particles, potentially containing *B. pseudomallei* over distances. Climate change-induced alterations in vegetation cover might expose soil to rainfall and winds[61], impacting the bacterial spread. Additionally, deforestation can disrupt natural barriers like trees and shrubs, accelerating water runoff[60,61] and potentially facilitating the wide-ranging dissemination of *B. pseudomallei* during flood. Considering these dynamics, the strategy of bacterial persistence likely plays a pivotal role in its widespread dissemination within the region, thereby influencing disease prevalence. Therefore, an effective disease control strategy should integrate both environmental and clinical public health measures to effectively mitigate the impact of melioidosis.

## Methods

### Ethical statement and data collection
Our research received approval from the ethics committees of each of the nine study hospitals in northeast Thailand where *B. pseudomallei* samples were obtained[8], as well as from the Mahidol University Faculty of Tropical Medicine (MUTM 2015-002-001 and MUTM 2021-055-01). The *B. pseudomallei* samples were collected between July 2015 to December 2018, comprising 1265 clinical isolates from northeast Thailand. Additionally, we incorporated a contemporaneous dataset from Thailand and neighbouring regions[15,29–34], comprising 15 clinical and 111 environmental isolates sourced from previous publications. In total, 1391 *B. pseudomlalei* genomes were used in this study. Their metadata and accession numbers were documented in Supplementary Data 1.

The northeast Thailand collection was obtained from patients participated in our longitudinal cohort study[8]. The patients were from nine provinces including Udon Thani ($n = 230$), Mukdahan ($n = 198$), Roi Et ($n = 195$), Surin ($n = 170$), Nakhon Phanom ($n = 135$), Buriram ($n = 123$), Sisaket ($n = 107$), Khon Kaen ($n = 96$) and Mahasarakham ($n = 11$), who were admitted to nine hospitals included in our cohort. The isolates were obtained from various clinical samples, including blood (71.8%), pus (12.5%), sputum (9.9%), body fluid (3.5%), urine

(1.9%) and tissue (0.4%). As latent infection accounts for < 5% of the cases, the majority of clinical cases likely directly acquired from the environment[8]. The numbers of enrolled cases was lower at the beginning of the study due to the delayed sample collection in some study sites, resulting in inconsistent number of bacterial isolates used across different sites. When applicable, a permutation test was performed to ensure that an unequal number of isolates did not impact the temporal or spatial analysis.

### Culture confirmation of *B. pseudomallei*, DNA extraction and whole genome sequencing
All 1265 *B. pseudomallei* samples from the northeast Thailand collection were cultured on Ashdown, selective agar plates and confirmed the species using latex agglutination test and matrix-laser absorption ionisation mass spectrometry (MALDI-TOF MS). A single colony from Ashdown agar plate was subjected to culture in Luria-Bertani (LB) broth (catalogue number 1551, Condalab, Spain) and subsequently used for DNA extraction. Genomic DNA was extracted using QIAamp DNA Mini Kit (catalogue number 51304, Qiagen, Germany). All genomic DNA were processed for the 150-base-read library preparation and sequencing using Illumina HiSeq2000 system with 100-cycle paired-end runs at Wellcome Sanger Institute, Cambridge UK. An average of 71X read depth was achieved. To control the potential contamination in each sample with other closely related species, we assigned taxonomic identity using Kraken[62] v.1.1.1. We then estimated the genome completeness and species confirmation using CheckM[63] v.1.2.2 and FastANI[64] v.1.31, respectively. The quality control data of the 1265 genomes were listed in Supplementary Data 2.

### Genome assembly and mapping from short read data
Short reads were de novo assembled using Velvet v.1.2.10[65], followed by optimisation and scaffolding to generate scaffolded contigs (see full protocol[29]). Data quality control is reported in Supplementary Data 2. Short reads were also mapped against several reference genomes, including a strain K96243[30] (accession numbers BX571965 and BX571966) to determine the whole population structure, and lineage-specific references to improve the resolution for lineage-specific analyses. We selected K96243[30] genome as a population-wide reference due to its origin in northeast Thailand, aligning with the geographical focus of our study. Additionally, its well-characterised and complete genome further streamlined subsequent analyses. For all mapping, variants were called using Snippy v.4.6.0 (https://github.com/tseemann/snippy). To avoid mapping errors and false SNPs, we filtered out SNPs covered by less than 10 reads and found in a frequency of less than 0.9.

### Defining whole population structure
**PopPUNK clustering.** PopPUNK v.2.6.0[35] was run on 1391 assembled genomes. To define the core and accessory distance between each pair of isolates, the assemblies were hashed at different k-mers. Several kmer comparison options (--k-step) and model fit options (--K) were tested to identify the parameters with the optimal fit based on density, transitivity and network scores. The optimal population model was fit using command line "poppunk --fit-model ----output < database > --min-k 15 --max-kmer 31 --max-a-dist 0.53 --K 4 --k-step 2". The model had a density of 0.028, a transitivity of 0.992, and a network score of 0.8961.

### Maximum likelihood phylogenies from core genome SNP, cgMLST and MLST
An alignment of full genome was created by mapping whole genome sequences of each *B. pseudomallei* against a complete genome of K96243[30] strain. From this alignment, 4221 cgMLST loci based on a scheme described in[37] were extracted and concatenated to form cgMLST alignment. Additionally, seven MLST loci, as per scheme

described in[38] were extracted from the same alignment and concatenated to create the MLST alignment. Core genome SNP alignment was identified from a full genome alignment using snp-sites[66] v.2.5.1, with genomic islands[51] masked. Separate maximum likelihood phylogenies were constructed for core genome SNP alignment, cgMLST alignment, and MLST alignment using IQ-TREE[36] v.2.0.3. Standard model selection in IQ-TREE determined the best-fit model as TVM + F + ASC + R6 for all three phylogenies. To access the robustness of the phylogenetic trees, a 1000 bootstrap support was performed for each tree.

### Comparison of population structure outlined phylogeny constructed from core genome SNPs, cgMLST, MLST and PopPUNK

To test for consistency between phylogenetic trees constructed from core genome SNPs, cgMLST and MLST alignment, we used the R package treespace[67] v. 1.1.4.3 to explore the tree tip distributions. We compared pairwise tree distances within the first 100 bootstraps within each alignment category (indicative of bootstrap support strength), and across trees generated from different alignment categories (indicating proximity between tree categories). The tree pairwise distances were computed, and principal components (PCs) were derived with eigenvalues calculated for different PCs. The similarity among phylogenies from each alignment category was assessed using two PCs dimensions, which jointly accounted for >90% of variability in pairwise distance (Supplementary Fig. 1a). The scatter plot of PCs revealed a close clustering of bootstrap trees from core genome SNP and cgMLST, while the bootstrap trees from MLST alignment showed greater dispersion, highlighting less consistency in the trees generated by the MLST approach. We further compared the consistency between the median phylogenetic tree of each alignment category and PopPUNK classification was visually compared using iTOL[68] (Supplementary Fig. 1b).

### Specific lineage analysis

To investigate the dissemination and genetic diversity within each dominant lineage, we conducted individual genome alignments, recombination removal, and maximum likelihood phylogeny. To enhance the sensitivity of variant, we selected closely related genomes as references for each lineage. Specifically, the complete genome *B. pseudomallei* strain K96243 (accession numbers BX571965 and BX571966) served as the mapping reference for lineage 1, while new reference genomes were created for lineages 2 and 3.

For lineage 2, we chose a representative isolate 27035_8#57 and subjected it to long-read sequencing on a local MinION sequencer following the manufacturer's standard protocol (Oxford Nanopore Technologies, Oxford, United Kingdom). A complete hybrid assembly of the long-read and short-read sequence data of this strain was performed using Unicycler[69] v.0.8.4. The resulting hybrid assembly of the 27035_8#57 genome was employed as the mapping reference for this lineage.

In the absence of representative complete genomes for lineage 3, we selected the best quality de novo assembly of isolate 27035_8#119 and orientated its contigs according to strain K96243 using ABACAS[70] v.1.3.1. This genome was used as a mapping reference for lineage 3.

For lineage-specific mapping, Snippy v.4.6.0 was employed as in the whole population analysis. All genome alignments were subjected to Gubbins[71] v.3.1.3, a recombination identification tool, to detect and remove recombination fragments. This process determined the genetic diversity introduced by horizontally acquired elements and vertically inherited SNPs, thereby producing recombination-free SNP alignments for phylogenetic reconstruction. Maximum-likelihood phylogenies were constructed using recombination-free SNP alignment of each dominant lineage using IQ-TREE[36] v.2.0.3 with TVM + F + ASC + R6 and 1000 replicates of bootstrap support. The overall proportion of nodes with ≥80% bootstrap support of lineage-specific phylogenies reached 83.5%.

### Dating the timeline for lineages and sub-lineages

To enable dating analysis, we further divided each lineage into sub-lineages using R package rhierbaps[72] v.1.1.4. We inferred evolutionary timeline and estimated the age of each lineage and sub-lineage based on the isolate's collection date. A Bayesian molecular dating provided in R package BactDating[73] v.1.1.1. was employed to assess the temporal signals by examining a positive correlation between the isolate's collection date and the root-to-tip distance. Recombination removed phylogenies were used in this analysis. A date-randomisation test, consisting of 100 permutations, was performed to assess the robustness of the temporal signal compared to noise.

Notably, the temporal signals were discernable at the sub-lineage level rather than the broader lineage level. Among the 10 sub-lineages, only one sub-lineage (lineage 1.3) exhibited a positive correlation in their clock signals. Given the limited sample size of lineage 1.3, we employed a strict clock model to prevent parameter over-fitting. We ran three independent Markov chain Monte Carlo (MCMC) chains, each spanning at least 100 million iterations, and sampled every 10,000 steps. The prior mutation rate derived from Pearson and colleagues[53] was used. Visual inspection of the trace from each MCMC chain confirmed signal convergence, with effective sampling size values > 200 for key parameters. Visualisation of results was performed using the R package ggtree[74] v.3.10.0 to generate credibility time-calibrated phylogeny for sub-lineage 1.3 (Supplementary Figs. 3c).

### Ancestral state reconstruction analysis

Ancestral trait reconstruction was conducted to discern the dissemination patterns of *B. pseudomallei* among provinces in northeast Thailand, focusing on dominant lineages. Due to varying number of isolates among provinces, the analysis excluded Mahasarakham, which had a limited dataset (*n* = 11), resulting in the analysis of eight provinces: Buriram, Khon Kaen, Mukdahan, Nakhon Phanom, Roi Et, Sisaket, Surin, and Udon Thani. This approach yielded 28 potential province-to-province transmission combinations. To mitigate sampling biases, we sub-sampled the phylogeny of each dominant lineage to have an equal number of isolates per province (*n* = 15 isolates) and permuted 1,000 times. Using the stochastic character mapping function (*make.simmap*) from the R package phytools[75] v.1.9.16, we conducted 100 simulations (nsim = 100) to reconstruct the provincial origins at each node in the sub-sampled phylogeny (1000 phylogenies per lineage). This allowed us to quantify transition events (Markov jumps) between province pairs and determine the cumulative branch length associated with each province (Markov rewards). A Mann–Whitney *U* test, with Bonferroni correction for multiple comparisons was applied to compare the transition event counts among provinces (Supplementary Fig. 6).

### Pan-genome analysis

All the study genomes were annotated using Prokka[76] v.1.14.5, and further used in the pan-genome analysis. Each genome has a median of 5845 coding sequences (CDS) predicted onto each genome with a range of 5642 to 6142 CDS per genome. Panaroo[77] v.1.3.3 was employed to estimate the pan-genome with a sensitive option and a cut-off sequence identity of 92% derived from the previous study[15]. The number of estimated genes falls within a comparable range to previous studies from a single population[29,53].

### Identification of dominant lineage-specific genes

We determined lineage-specific genes by assessing their prevalence within dominant lineage or any of their sub-lineages, requiring a high occurrence (95%) within these specific groups while maintaining a low presence in non-dominant lineages. To achieve this, three thresholds were employed: strict (95% occurrence in dominants vs 5% occurrence in non-dominants), intermediate (95% in dominants vs 10% in

non-dominants), and relaxed (95% in dominants vs 15% in non-dominants). Based on a visual examination of gene distribution patterns (Fig. 3a, Supplementary Fig. 7), the relaxed threshold was used to maximise the number of genes included in subsequent analysis.

### Identification of gene ontology (GO: terms)
Amino acid sequences of lineage-specific genes were submitted to InterPro database[46] (https://www.ebi.ac.uk/interpro/) which characterised the function of lineage-specific genes based on biological processes, molecular functions, and cellular compartments (Fig. 3b; Supplementary data 3).

### Transcriptomic analysis of dominant lineage-specific genes
**Lineage 1 transcriptome analysis.** The analysis focused on an expression profile of a strain K96243, which serves as a representative of lineage 1. Data was sourced from microarray experiment generated by Ooi et al. [47] and accessed through the Gene Expression Omnibus (GEO) under accession number GSE43205.

To understand the difference between environmental and infection conditions, we compared the expression profile of K96243 being exposed to water and K96243 recovered from infected mice. To simulate environmental conditions, K96243 was cultured to log phase in LB medium, subsequently washed with sterile deionising water, and suspended in water. To emulate infection conditions, BALB mice were infected with 1000 CFU of *B. pseudomallei*, and bacteria were harvested from the lungs three days post-infection. Two replicates were performed for each condition. We retrieved microarray data from GEO using the R package GEOquery[78] v.2.58.0, and differential gene expression analysis was performed using the R package limma[79] v.3.58.1.

We used binary expression patterns reported in Ooi et al. [47] to compare the expression profiles of lineage-specific genes against the remaining genes in K96243 across 62 conditions. This enabled the comparison of the count of expressed genes within the lineage-specific category against the remainder of the genes for each condition using Fisher's exact test, with multiple testing adjustments via Benjamini-Hochberg corrections.

**Lineage 2 transcriptome analysis.** This analysis focused on the expression profile of a strain UKMD286, representative of lineage 2. RNAseq data was obtained from an experiment conducted by Ghazali et al. [48] and accessed through the European Nucleotide Archive (ENA) (E-MTAB-11200).

To simulate environmental conditions, UKMD286 was cultured in BHIB medium overnight, resuspended, and inoculated into soil extract medium. For infection conditions, BALB mice were infected with UKMD286, and the bacteria were harvested from spleens five days post-infection. Each experiment was conducted with two replicates. FastQC v.0.11.9 and FastXtool v.0.0.14 were used to pre-processed sequenced reads. Raw reads were aligned to UKDM286 genome using Hisat2[80] v. 2.2.1 with differential gene expression performed using the R package DESeq2[81] v.1.40.2.

**Lineage 3 transcriptome analysis.** We used a strain UKMH10 to represent the expression profile of lineage 3. Data was originated from an RNAseq experiment conducted by Kong et al. [49] and was accessed through the European Nucleotide Archive (ENA) (PRJEB53338).

To replicate environmental conditions, UKMH10 was cultured in LB medium overnight and subcultured into soil extract medium. To simulate infection conditions, UKMH10 was cultured in LB medium overnight and inoculated into human plasma, then incubated at 37 °C to mimic the human body temperature. Bacterial cells were harvested once the absorbance reading of the bacterial cultures at 600 nm (OD600) reached 0.5. Each experiment was performed with two replicates. FastQC v.0.11.9 and FastXtool v.0.0.14 were used to pre-

processed sequenced reads. Raw reads were aligned to UKMH10 genome using Hisat2[80] v.2.2.1 with differential gene expression performed using the R package DESeq2[81] v.1.40.2.

### Statistics and reproducibility
We did not use statistical methods to predetermine the sample size, as the number of isolates used depended on *B. pseudomallei* isolates obtained from melioidosis patients admitted to nine study hospitals in northeast Thailand during the study period. All newly sequenced *B. pseudomallei* genomes from northeast Thailand were combined with existing sequenced data from Southeast Asia to contextualise our findings. We used independent approaches to outline population structure into groups, ensuring reproducibility. Subgroup analyses focused on three dominant lineages, excluding smaller groups from the subsequent analyses. These analyses included reconstructing the geographical and temporal spread of each dominant lineage, identifying lineage-specific genes and their recombination patterns, and comparing the transcriptome profiles of lineage-specific genes in environmental and infection conditions. Date-randomisation permutation tests were performed to ensure that the detected temporal signals were not random. Additionally, permutations were conducted to reduce bias from the varying number of isolates from each province during the reconstruction of province-to-province dissemination history. The investigators were not blinded to allocation as the outcomes were objectively measured and did not require subjective interpretation.

### Reporting summary
Further information on research design is available in the Nature Portfolio Reporting Summary linked to this article.

### Data availability
The newly sequenced 1265 *B. pseusomallei* genomes from northeast Thailand generated in this study have been deposited in the European Nucleotide Archive (ENA) under study accession number PRJEB25606 and PRJEB35787. The accession numbers for individual genomes are provided in Supplementary Data 1. We sourced existing RNA data used to compare gene expression during infection and in the environment for representative strains of lineages 1, 2, and 3 from the following repositories: NCBI Gene Expression Omnibus (GEO) under accession number GSE43205 (lineage 1) [https://www.ncbi.nlm.nih.gov/geo/query/acc.cgi?acc=GSE43205], and the ENA under accession numbers E-MTAB-11200 (lineage 2) [https://www.ebi.ac.uk/biostudies/arrayexpress/studies/E-MTAB-11200] and PRJEB53338 (lineage 3) [[https://www.ncbi.nlm.nih.gov/bioproject/?term=PRJEB53338]. Source data are provided with this paper. There is no restriction on data availability. Source data are provided with this paper.

### Code availability
All software utilised for data analysis is open source.

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

## Acknowledgements

We would like to thank physician and microbiology staff at the Udon Thani Hospital, Khon Kaen Hospital, Srinakarin Hospital, Nakhon Phanom Hospital, Mukdahan Hospital, Roi Et Hospital, Surin Hospital, Sisaket Hospital and Buriram Hospital for reporting the case and providing the facilities for bacterial collection. We sincerely appreciate the transcriptomic data of B. *pseudomallei* UKMD286 and UKMH10 provided by Professor Sheila Nathan and colleagues. We are grateful to Dr. Soe Htet Aung for supporting geographical map plot. This work was funded by Mahidol University (MU-KMUTT Biomedical engineering and Biomaterials Consortium) and Royal Golden Jubilee Ph.D. Programme (RGJ-ASEAN) (http://rgj.trf.or.th) through RS and NC. CCho was funded by Wellcome International Master Fellowship (221418/Z/20/Z) and Prince Mahidol PhD studentship. CChe was funded by Wellcome International Intermediate Fellowship (216457/Z/19/Z), Sanger International Fellowship, and Thailand Health System Research Institute (67-138). NC and TEW were supported by the National Institute of Allergy and Infectious Diseases of the National Institutes of Health (NIAID/NIH) (https://www.nih.gov) under Award Number U01AI115520. The content is solely the responsibility of the authors and does not necessarily represent the official views of the National Institutes of Health. This research was funded in part, by the Wellcome Trust [220211 and 206194]. For the purpose of Open Access, the author has applied a CC BY public copyright license to any Author Accepted Manuscript version arising from this submission. The funders had no role in study design, data collection and analysis, decision to publish, or preparation of the manuscript.

## Author contributions

NC and CChe conceived and designed the study, as well as administered and supervised the project. NC secured funding to collect isolates, while CChe obtained funding for sequencing and downstream analysis. NC, RS, TEW, and NS collected and identified bacterial isolates, with NC, RS, ST, and RP gathering and curating clinical data. CChe, JP, NRT, performed

short-read sequencing while NC, RS, JT, EB and WC performed long-read sequencing. NPJD and NC provided reagents. NRT and JP contributed software tools. RS, CCho, and CChe performed bioinformatics analyses. CChe, NC, NRT and JP interpret the analyses. The original draft was written by RS, NC, and CChe. CChe reviewed, edited, and wrote revision drafts. All authors read and approved the manuscript.

## Competing interests
The authors declare no competing interests.

## Additional information

[1]Department of Microbiology and Immunology, Faculty of Tropical Medicine, Mahidol University, Bangkok, Thailand. [2]Mahidol Oxford Tropical Medicine Research Unit, Faculty of Tropical Medicine, Mahidol University, Bangkok, Thailand. [3]Department of Veterinary Medicine, University of Cambridge, Cambridge, UK. [4]Department of Clinical Tropical Medicine, Faculty of Tropical Medicine, Mahidol University, Bangkok, Thailand. [5]Centre for Tropical Medicine and Global Health, Nuffield Department of Medicine, University of Oxford, Oxford, UK. [6]Center for Medical Genomics, Faculty of Medicine Ramathibodi Hospital, Mahidol University, Bangkok, Thailand. [7]Division of Pulmonary, Critical Care & Sleep Medicine, Department of Medicine, University of Washington, Seattle, WA, USA. [8]Department of Global Health, University of Washington, Seattle, WA, USA. [9]Parasites and Microbes Wellcome Sanger Institute, Cambridge, UK. [10]Faculty of Infectious and Tropical Diseases, London School of Hygiene & Tropical Medicine, London, UK. [11]Previous Affiliations: Bioinformatics and Systems Biology Program, School of Bioresource and Technology, King Mongkut University of Technology Thonburi, Bangkok, Thailand. [12]These authors contributed equally: Claire Chewapreecha, Narisara Chantratita. ✉e-mail: claire@tropmedres.ac; narisara@tropmedres.ac

