## [Peer Review File · Nature Communications]

REVIEWER COMMENTS

Reviewer #1 (Remarks to the Author):

NCOMMS-23-24337

Genetic diversity, determinants, and dissemination of *Burkholderia pseudomallei* lineages implicated in melioidosis in northeast Thailand

Reviewer comments

Manuscript Overview

Very few studies have been published on large clinical cohorts of melioidosis – probably the most well-known clinical cohort is the “32-year Darwin melioidosis study” in Darwin Australia. This study by Seng et al is an extremely valuable study that is the first large clinical cohort that has been presented together with comprehensive genomics to be described outside of Australia and will be extremely valuable to the global melioidosis community and the Thai community. Here the authors describe a comprehensive population genomics analysis of ~1300 isolates from northeast Thailand and transcriptome analysis of one lineage, all methods used for the analysis are sound.

Minor comments/changes:

- Overall - this paper is exciting, novel and has been well written and thought-out.
- In the methods section please add a section on how the cultures were confirmed to be *B. pseudomallei* and how the isolates were cultured, under what conditions and indicate if sequencing was performed on DNA obtained from single colonies.
- In the methods section Line 328 please indicate what quality metrics your assemblies had – what was your cutoff used to deem an assembly to be good quality and hence used in analyses? Please list your contig no and N50 cutoffs for this.
- Does your lineage classification hold and remain consistent when additional global genomes are added to the analysis? Or does it change when genomes are removed or added? Might be worth mentioning the limitations to using this to classify lineages.

- Please justify for the phylogenetic tree that contains all isolates - why you used K96243 as the reference? Is this because it's the most ancestral Asian *B. pseudomallei* isolate? Or because it's a high quality well known characterised *B. pseudomallei* closed genome?
- The paper is well written but just some minor spelling errors - so suggest reading throughout to scan for these and correct e.g. line 369 "was used a reference sequence" – should be "was used as a reference sequence" and line 415 "low frequency may not specific to dominant lineages" should be "low frequency may not be specific to dominant lineages"
- In the methods section line 437 "Transcriptomic analysis of dominant lineage-specific genes and SNPs" – This section I think would be hard to replicate based on the information provided in the methods section. Please provide more details on how the transcriptome analysis was undertaken so that it is clearer to readers – in the instance that this be replicated.
- Transcriptome analysis was only done for one lineage (k96243 lineage 3) – this type of experiment would be very useful to replicate for representative strains belonging to each lineage – is this something the authors are considering?
- The majority of the discussion section does focus on discussing the transcriptional results for Lineage 3 based on the single experiment with K96243 – I think the discussion could be made much more dynamic if there was also a focus on the genetic diversity and population structure results which is the main component of this paper – please add a paragraph to the discussion on discussing the lineage and ST trends that were noted among the patient isolates – do you see big temporal and spatial shifts in STs and lineages? – would this have implications on public health surveillance in the area? Any introductions from other nearby countries (shared STs)? and do trends you see in Thailand contrast or agree with those seen in other endemic regions including Australia. Further the authors have some very interesting results in terms of *B. pseudomallei* dispersal and weather patterns – this is also not mentioned or discussed in the discussion section which is a missed opportunity – please add a section discussing these important results and how they compare to other endemic countries.
- For Supplementary figure S1 - I'd suggest changing the colour scheme for the "year" as the colour pallet for "year" is very similar to "province" and this makes it difficult to interpret the coloured strips. Additionally, the node labels are illegible – if they are required to be read, I suggest making the figures larger - if the node labels are not required then I'd suggest removing them from all four phylogenetic trees. Similarly, Figure 2 the colour pallets used across the colour strips are very similar and make it hard for the reader to interpret the tree – would suggest choosing different colours so that each strip represent a different pallet.

Reviewer #2 (Remarks to the Author):

The paper “Genetic diversity, determinants, and dissemination of *Burkholderia pseudomallei* lineages implicated in melioidosis in northeast Thailand” is a genomic analysis of 1265 *B. pseudomallei* isolates from melioidosis patients. The paper is well written and does present some interesting data. It is an impressive collection of strains and sequence data, however, the data and work only partially satisfy the findings. Unless more analyses are done the significance is diminished. *B. pseudomallei* is an environmentally mediated pathogen and the authors attempt to make many conclusions about transmission of the disease without bringing the important component of the reservoir into the research. Many of the transcriptional and metabolic aspects of the investigation do not mesh well with the genomic or epidemiological analyses. It is unclear why only type strain K96243 (as a lineage 3 model) is the only strain used in the transcriptional experiments. Findings would be much stronger using a selection of clinical strains sequenced in this work and comparing expression of common genes between them under the different conditions. The transcriptional findings suggest a trend in the lineage 3 model strains K96243 that is not well supported by statistical analyses. The epidemiological conjecture would benefit greatly from a more intensive geospatial analyses to bolster support for lineage specific transmission across this region of Thailand.

The main issue is that one cannot make firm conclusions about transmission of an environmentally mediated pathogen based on clinical isolates alone since human-human transmission is extremely rare. It is known that *B. pseudomallei* has a high diversity in the soil, with a potential for numerous STs to be isolated from the same soil sample. Which ST infects a person may be totally random. The random hypothesis is supported by the ML SNP tree in that there are no discernable patterns of ST, year, or province. Pulling STs, cgMLST, or sequence data from environmental isolates across northeast Thailand over the same time-period or even from a historical perspective would strengthen the findings and might support the notion of directional dispersal of strains across Thailand rather than long term complex reservoir. Indeed, one could also examine predominant STs from bordering countries to understand the movement of strains into bordering Thai provinces.

Specific Comments:

Line 39: affect 165,000 people globally per year, of which an estimated 89,000 are fatal.

Line 54: Streams instead of water streams

Line 59: Previous studies have

Line 62: disease severity ??

Line 64: only clinical isolates? And then only one type strain investigated for strategies of survival.

Line 68: representative

Line 73: The meaning of success is confusing here. Does success mean fitness? Ability to infect? To this reviewer the human infection is accidental and survival in the environment (aka soil, water, streams) would be a better measure of success. An ST that is successful at survival in the soil may not be infectious or virulent. Without environmental strain analysis you can't determine that.

Line 79: It is unclear what determining lineages “by size” means.

Line 90: STs or cgMLSTs, the authors should consider looking at these strains in terms of cgMLST profiles or wgMLST. These classification schemes can be downsampled to look at lower resolution population structures as well as high resolution.

Line 96: Dissemination route used in this fashion is highly confusing since the person-person movement networks aren't analyzed and again there are no matching environmental samples.

Line 106: The comment about bidirectional transmission events with similar number of transmission events acting in opposite directions is not only confusing but could indicate all STs are present in the environment.

Line 123-129: These statements are highly speculative given the amounts and type of data presented.

Line 193: Provide data or references for the association of genetic elements like IS elements and bacteriophages with GIs.

Line 227: The authors focus on functions of RM systems as potential competitive advantage in strains of certain lineages then offer speculation that it impacts the genetic landscape of *B. pseudomallei* in NE Thailand. This hypothesis can be demonstrated through some genetic experiments and would increase the significance of this work.

Line 238: , a hyperendemic melioidosis region in Asia.

Line 243: The authors anticipate human melioidosis cases indicate environmental prevalence. This needs to be demonstrated not merely anticipated to explain significance of the findings.

Line 251-253: Without an environmental component to the study this cannot be known.

Line 264-266: A statement of limitation of findings to lineage 3 is mentioned, the data need not be limited to lineage 3. You have numerous strains to make numerous conclusions.

Line 285: Is a host-infection an evolutionary dead end for *B. pseudomallei*? Take glanders for example. It used to be more similar to *B. pseudomallei* and over time became host-adapted. The soil and water environment is full of diverse hosts, from single to multicellular organisms, that have no doubt influenced the evolution of *B. pseudomallei*. Even so far as shaped its very genome with selective acquisition of virulence factors that allow its survival in diverse environments.

Reviewer #3 (Remarks to the Author):

Seng and co-workers have carried out a deep genomic study of a lethal opportunistic human pathogen, *Burkholderia pseudomallei* (Bps). The bacterium, Bps, has highly recalcitrant, environmental lifestyle and with rapid climatic changes, understanding its dual lifestyle is need of the hour. The samples are from an

hyperendemic endemic collected over period of four years that is affected by typhoons, monsoon directions, and other climatic alterations through river flow and canal systems. Through deep genomics study of more than 1000 samples, the authors have identified four dominant lineages and tried to understand their success and transmission through further phylogenetic analysis of dominant lineages, comparative genomic studies involving identifying genes harbouring SNPs and genes unique to the dominant lineages. The authors report the role of horizontal gene transfer in acquisition of new genes for adaptation in extreme environments that is encountered by Bps and recombination in acquisition of new mutations and also genes in the dominant lineage(s). In one of the dominant lineages, the authors also provide evidence through transcriptomic studies that the lineage specific genes harbouring variations or unique genes are important for adaption/fitness rather than virulence/infections indicating environmental persistence in success and transmission route of these dominant lineage(s). Accordingly, the authors also correlate the dominance and movement of lineages according to climatic conditions prevailing in the diverse geographic regions of the country.

Overall, the study is a major and novel effort in understanding a fatal pathogen in general and Bps in particular. However below are the major comments that author need to address before any decision is taken.

1) The coverage of genomes generated in the study is just below 100X. Also, authors need to provide genomic completeness/contamination report as generated by software's like CheckM.

2) NCBI Genbank accession numbers with annotation of the genome is missing. Just SRA data will not be helpful for the researchers who are not good at bioinformatics and who work on microbiology, genetics, and physiology of the organism.

3) Have they confirmed the all the isolates are *B. pseudomallei* by ANI cutoff with the type strain?

4) The authors need to work on the title and rest of the manuscript as the findings related to genes unique or under variation in the dominant lineages. Sometime it sounds that the authors are reporting genes unique or under variation in the each or every of the dominant lineages or between two dominant lineages. Hence authors need to clarify the same. For example, Distinct gene content and SNPs described only to 1, 2 lineages ...no inform on such results with other lineages...

5) While the phylogenomic evidence for dominant lineages is clear, the findings related to unique genes and lineage specific genes carrying SNPs is from one particular dominant lineage, while transcriptomic studies are from another dominant lineage and mapping to a genomic island from another dominant lineages. While it is important to understand the evolution and transmission as revealed by dominant lineages, the authors must not end up confusing. Hence there is need to make the results/discussion clear.

6) Throughout the sampling period, the four dominant lineages consistently dominated in terms of size and persistence, indicating their robust and stable presence in the sampled population (Fig. 1d).” Authors need to provide year wise phylogenetic trees (may be as supplementary) to indicate the presence of these dominant lineages as the Fig. 1d is not clear.

7) Also, the authors can clarify the year wise transmission pattern of dominant lineages to see if their hypothesis based on all the isolates across year holds true.

8) Similarly, recombination studies, mapping of the unique genes/lineage specific genes in dominant lineages in each year is something missing in the study.

9) Also their mention and generation of complete genome sequence (again in one or two dominant lineage) is not clear and incomplete. Also authors need to provide details regarding obtaining complete genome sequence using nanopore data.

10) Overall, the authors have planned an important and innovative study in a lethal pathogen, however there is scope to add on with further more analysis possible with their data. With the kind of year wise resource and data, the authors have not completely justified with analysis and sounds at many places incomplete/abrupt.

RE: NCOMMS-23-24337

Genetic diversity, determinants, and dissemination of *Burkholderia pseudomallei* lineages implicated in melioidosis in northeast Thailand

Reviewer comments

Reviewer #1 (Remarks to the Author):
Manuscript Overview

Very few studies have been published on large clinical cohorts of melioidosis – probably the most well-known clinical cohort is the “32-year Darwin melioidosis study” in Darwin Australia. This study by Seng et al is an extremely valuable study that is the first large clinical cohort that has been presented together with comprehensive genomics to be described outside of Australia and will be extremely valuable to the global melioidosis community and the Thai community. Here the authors describe a comprehensive population genomics analysis of ~1300 isolates from northeast Thailand and transcriptome analysis of one lineage, all methods used for the analysis are sound.

We are pleased to contribute the geographical representation of the melioidosis cohort, and deeply thankful for the encouraging feedback and constructive comments. They have been instrumental in strengthening this manuscript. All comments have been incorporated into the revised manuscript, marked in the “manuscript with marked changes” file.

Minor comments/changes:

1. Overall - this paper is exciting, novel and has been well written and thought-out.

We are delighted to learn that the reviewer found the manuscript exciting.

2. In the methods section please add a section on how the cultures were confirmed to be *B. pseudomallei* and how the isolates were cultured, under what conditions and indicate if sequencing was performed on DNA obtained from single colonies.

We appreciate the importance of this information and apologise for leaving it out previously. We have expanded the methods section to include information about how we confirmed the presence of *B. pseudomallei* using biochemical tests, latex agglutination assay, and MALDI-TOF MS. Additionally, we have added information about the culture conditions and the use of single bacterial colonies for DNA sequencing as suggested (Line 381-385).

3. In the methods section Line 328 please indicate what quality metrics your assemblies had – what was your cutoff used to deem an assembly to be good quality and hence used in analyses? Please list your contig no and N50 cutoffs for this.

We thank the reviewer for this comment. We employed CheckM to evaluate assembled genomes, with reported completeness ranging from 99.3 - 100.0% and contamination at 0.0 - 1.1%, ensuring high data quality. We have now incorporated a summary of assembled contigs, N50 value, genome completeness and contamination scores are now included in both the method section (line 390-392) and Supplementary data 2.

4. Does your lineage classification hold and remain consistent when additional global genomes are added to the analysis? Or does it change when genomes are removed or added? Might be worth mentioning the limitations to using this to classify lineages.

We deeply appreciate the reviewer’s insightful feedback. Lineage classification is dynamic and can change over time as the bacterium continue to evolve. To enhance robustness of our analysis, we now incorporated contemporaneous *B. pseudomallei* samples from diverse

regions in Thailand and neighbouring countries into the analysis. The results showed consistency with our previous findings, with minor lineages and singletons remaining distinct, but two dominant lineages merged, resulting in three dominant lineages instead of four. We acknowledge this limitation and emphasise how our results may vary with different input data in the discussion (Line 305-313).

5. Please justify for the phylogenetic tree that contains all isolates - why you used K96243 as the reference? Is this because it's the most ancestral Asian *B. pseudomallei* isolate? Or because it's a high quality well known characterised *B. pseudomallei* closed genome?

The selection of K96243 was based on several factors: firstly, its origin from northeast Thailand, which aligns with the geographical focus of our study; secondly, its complete genome, which allows for a more comprehensive analysis; and thirdly, its extensive characterisation in multiple studies, making it a reliable reference. In response to the reviewer's comment, we have included the rationale for using K96243 in the methods section (Line 399-402).

6. The paper is well written but just some minor spelling errors - so suggest reading throughout to scan for these and correct e.g. line 369 "was used a reference sequence" – should be "was used as a reference sequence" and line 415 "low frequency may not specific to dominant lineages" should be "low frequency may not be specific to dominant lineages"

We are thankful to the reviewer's input and have made the suggested edits.

7. In the methods section line 437 "Transcriptomic analysis of dominant lineage-specific genes and SNPs" – This section I think would be hard to replicate based on the information provided in the methods section. Please provide more details on how the transcriptome analysis was undertaken so that it is clearer to readers – in the instance that this be replicated.

We appreciate the reviewer's comment and have substantially expanded the corresponding method section to ensure the reproducibility of our study (Line 536-577).

8. Transcriptome analysis was only done for one lineage (k96243 lineage 3) – this type of experiment would be very useful to replicate for representative strains belonging to each lineage – is this something the authors are considering?

In response to the reviewer's comment, we have now included the transcriptome data for representative strains for all dominant lineages. The new data included K96243 representing the new lineage 1 (merging former lineage 2 and lineage 3), UKMD286, representing new lineage 2; and UKMH10, representing new lineage 3. These new results further support previous findings from a single lineage, indicating that lineage-specific genes show elevated expression under environmental conditions. This underscores the importance of environmental survival in determining the success of for each lineage (Line 207-230).

9. The majority of the discussion section does focus on discussing the transcriptional results for Lineage 3 based on the single experiment with K96243 – I think the discussion could be made much more dynamic if their was also a focus on the genetic diversity and population structure results which is the main component of this paper – please add a paragraph to the discussion on discussing the lineage and ST trends that were noted among the patient isolates – do you see big temporal and spatial shifts in STs and lineages? – would this have implications on public health surveillance in the area? Any introductions from other nearby countries (shared STs)? and do trends you see in Thailand contrast or agree with those seen in other endemic regions including Australia. Further the authors have some very interesting results in terms of *B. pseudomallei* dispersal and weather patterns – this is also not mentioned

or discussed in the discussion section which is a missed opportunity – please add a section discussing these important results and how they compare to other endemic countries.

We appreciate the valuable input from the reviewer. In the revised discussion, we have now included a dedicated paragraph to address lineages and their temporal and spatial shift (Line 305-313). These additions significantly enhance the depth and scope of our discussion. Regarding the comparison with Australian isolates, we anticipate that a more comprehensive analysis will require a larger contemporary dataset, and we are eager to collaborate with the Australian teams for future investigations.

10. For Supplementary figure S1 - I'd suggest changing the colour scheme for the "year" as the colour pallet for "year" is very similar to "province" and this makes it difficult to interpret the coloured strips. Additionally, the node labels are illegible – if they are required to be read, I suggest making the figures larger - if the node labels are not required then I suggest removing them from all four phylogenetic trees. Similarly, Figure 2 the colour pallets used across the colour strips are very similar and make it hard for the reader to interpret the tree – would suggest choosing different colours so that each strip represent a different pallet.

We are thankful to the reviewer comments and have generated new figures and supplementary figures to be more visually informative. Again, we thank the reviewer for their thoughtful and constructive comments, which have enriched the quality and depth of our work.

Reviewer #2 (Remarks to the Author):

The paper "Genetic diversity, determinants, and dissemination of *Burkholderia pseudomallei* lineages implicated in melioidosis in northeast Thailand" is a genomic analysis of 1265 *B. pseudomallei* isolates from melioidosis patients. The paper is well written and does present some interesting data. It is an impressive collection of strains and sequence data, however, the data and work only partially satisfy the findings. Unless more analyses are done the significance is diminished. *B. pseudomallei* is an environmentally mediated pathogen and the authors attempt to make many conclusions about transmission of the disease without bringing the important component of the reservoir into the research. Many of the transcriptional and metabolic aspects of the investigation do not mesh well with the genomic or epidemiological analyses. It is unclear why only type strain K96243 (as a lineage 3 model) is the only strain used in the transcriptional experiments. Findings would be much stronger using a selection of clinical strains sequenced in this work and comparing expression of common genes between them under the different conditions. The transcriptional findings suggest a trend in the lineage 3 model strains K96243 that is not well supported by statistical analyses.

We genuinely appreciate the reviewer's interest in the data and the valuable feedback provided. The manuscript has undergone substantial revisions, as demonstrated in the "manuscript with marked changes" file.

Despite the limited availability of environmental samples, we have now incorporated contemporaneous environmental samples from Thailand and neighbouring countries. We confirm that both environmental and clinical samples share lineage membership, reinforcing the prior knowledge that patient isolates originate from environmental reservoirs and that signals related to bacterium survival under environmental pressure can be detected in clinical isolate genomes.

Furthermore, we have extended the transcriptome analysis to include representative strains from additional dominant lineages: K96243 (new lineage 1), UKMD286 (new lineage 2), and

UKMH10 (new lineage 3). These results corroborate the earlier findings within a single lineage, demonstrating that lineage-specific genes exhibit elevated expression under environmental conditions. By incorporating these transcriptome models, our statistical analysis is now more robust, offering stronger support for the role of environmental adaptation in shaping the success of each lineage.

The epidemiological conjecture would benefit greatly from a more intensive geospatial analyses to bolster support for lineage specific transmission across this region of Thailand.

We agree with the reviewer and acknowledge a gap in conducting a more intensive geospatial analyses with patient movement data combined. Unfortunately, we did not obtain patient consent to collect this type of data, which restricts us from conducting analyses at a greater resolution than what is presented in the current study. A reconsenting process would be necessary to pursue more in-depth analyses in the future.

The main issue is that one cannot make firm conclusions about transmission of an environmentally mediated pathogen based on clinical isolates alone since human-human transmission is extremely rare. It is known that *B. pseudomallei* has a high diversity in the soil, with a potential for numerous STs to be isolated from the same soil sample. Which ST infects a person may be totally random. The random hypothesis is supported by the ML SNP tree in that there are no discernable patterns of ST, year, or province. Pulling STs, cgMLST, or sequence data from environmental isolates across northeast Thailand over the same time-period or even from a historical perspective would strengthen the findings and might support the notion of directional dispersal of strains across Thailand rather than long term complex reservoir. Indeed, one could also examine predominant STs from bordering countries to understand the movement of strains into bordering Thai provinces.

We appreciate the reviewer's comment and acknowledge that due to the scarcity of the environmental isolates, we could not provide a comprehensive report on bacterial genetic diversity in the environment. Nevertheless, we have now demonstrated shared lineage membership between patients and environmental isolates, highlighting their shared genetic similarity. With careful interpretation, the conclusion drawn is that clinical isolates can be used to explain *B. pseudomallei* transmission patterns in the environment.

However, we respectfully disagree with the suggestion to use STs or cgMLST for transmission analysis. These methods have limited resolution and cannot provide the depth required for transmission analysis. To reliably link descendants to ancestors within lineages or sub-lineages, we need whole-genome sequenced data to differentiate vertically inherited single nucleotide polymorphisms (SNPs) from horizontally acquired ones. We kindly request to omit the use of STs or cgMLST for linking environmental to clinical samples, or tracing the transmission patterns, as we believe this will weaken our analyses.

Nonetheless, we agree with the reviewer that cgMLST may still serve as valuable tools to approximate the population structure, and we thoroughly explored this in a separate analysis (Supplementary Figure 1).

Specific Comments:

Line 39: affect 165,000 people globally per year, of which an estimated 89,000 are fatal.

This has been edited accordingly (Line 51-52)

Line 54: Streams instead of water streams

This has been edited accordingly (Line 69)

Line 59: Previous studies have

This was removed during revision.

Line 62: disease severity ??

This was removed during revision.

Line 64: only clinical isolates? And then only one type strain investigated for strategies of survival.

We have now included the environmental isolates and additional transcriptome models across all dominant lineages to investigate the bacterial survival strategies (Line 207-230)

Line 68: representative

This has been edited accordingly (Line 88)

Line 73: The meaning of success is confusing here. Does success mean fitness? Ability to infect? To this reviewer the human infection is accidental and survival in the environment (aka soil, water, streams) would be a better measure of success. An ST that is successful at survival in the soil may not be infectious or virulent. Without environmental strain analysis you can't determine that.

We thank the reviewer for this comment and have incorporated additional environmental isolates to strengthen the analysis. We fully agree that survival in the environmental serves as a more meaningful measure of success and have edited the text accordingly (Line 185-187). Our new analysis further supports this notion, revealing elevated expression patterns of dominant lineage-specific genes even in clinical strains under environmental conditions, highlighting their significance in the environment. This aligns with the reviewer's suggestion that environmental survival is a crucial factor for *B. pseudomallei*'s success as a pathogen.

Line 79: It is unclear what determining lineages "by size" means.

This was removed during revision.

Line 90: STs or cgMLSTs, the authors should consider looking at these strains in terms of cgMLST profiles or wgMLST. These classification schemes can be downsampled to look at lower resolution population structures as well as high resolution.

We thank the reviewer for this comment and have addressed it by conducting a separate analysis on MLST and cgMLST. While their resolution is insufficient for transmission analysis, we have demonstrated their utility as Supplementary Figure 1.

Line 96: Dissemination route used in this fashion is highly confusing since the person-person movement networks aren't analyzed and again there are no matching environmental samples.

Due to ethical consideration, we were unable to access data on person-to-person movement for this study. This limits the scope of our study beyond the current analysis. However, the inclusion of additional environmental samples, as suggested by the reviewer, has reinforced our previous findings. We believe that these findings still provide valuable insights into the patterns of *B. pseudomallei* dissemination.

Line 106: The comment about bidirectional transmission events with similar number of transmission events acting in opposite directions is not only confusing but could indicate all STs are present in the environment.

We apologise for any confusion caused by our initial communication and have revised the text to improve clarity. The transmission analysis did not suggest that all lineages (or STs) are present in the environment (as all clinical isolates came from the environment); rather, they are distributed across various geographical regions, with varying transmission patterns in specific directions.

Line 123-129: These statements are highly speculative given the amounts and type of data presented.

We have now included the environmental isolates to substantiate our use of the clinical isolates as a proxy to identify dissemination patterns (Line 120-129, and Line 301-205).

Line 193: Provide data or references for the association of genetic elements like IS elements and bacteriophages with GIs.

We thank the reviewer for this input and have provided additional references to support the arguments (Line 252).

Line 227: The authors focus on functions of RM systems as potential competitive advantage in strains of certain lineages then offer speculation that it impacts the genetic landscape of *B. pseudomallei* in NE Thailand. This hypothesis can be demonstrated through some genetic experiments and would increase the significance of this work.

We are grateful for the reviewer comment and intrigued to explore the suggested idea. However, we have to reserve this for future work. Our study timeframe is limited by the progress of our PhD student, Rathenin Seng, who needs to graduate within a specified timeline. Nevertheless, we are open to exploring this avenue in future research.

Line 238: , a hyperendemic melioidosis region in Asia.

This has been edited accordingly (Line 292)

Line 243: The authors anticipate human melioidosis cases indicate environmental prevalence. This needs to be demonstrated not merely anticipated to explain significance of the findings.

We thank the reviewer for this comment. We have now included environmental isolates to support the genetic overlap between environmental and clinical samples. This reinforces the use of clinical isolates as a proxy for the environmental prevalence (Line 120-129, and Line 301-205)

Line 251-253: Without an environmental component to the study this cannot be known.

We have incorporated the environmental isolates as suggested by the reviewer, and carefully revised the text to reflect the new findings (Line 120-129, and Line 301-205).

Line 264-266: A statement of limitation of findings to lineage 3 is mentioned, the data need not be limited to lineage 3. You have numerous strains to make numerous conclusions.

We agree with the reviewer and have now included additional transcription model using representative strains from other lineages. We have revised the text to present these new findings and limitations associated with using these representative strains (Line 207-230 and Line 315-329).

Line 285: Is a host-infection an evolutionary dead end for *B. pseudomallei*? Take glanders for example. It used to be more similar to *B. pseudomallei* and over time became host-adapted. The soil and water environment is full of diverse hosts, from single to multicellular organisms, that have no doubt influenced the evolution of *B. pseudomallei*. Even so far as shaped its very genome with selective acquisition of virulence factors that allow its survival in diverse environments.

We agree with the reviewer and have refined the text to elucidate how environmental habitats contribute to shaping multiple selection pressures, resulting in the observed lineage-specific genes in this study. We greatly appreciate the comments and have incorporated them the revised manuscript.

Reviewer #3 (Remarks to the Author):

Seng and co-workers have carried out a deep genomic study of a lethal opportunistic human pathogen, *Burkholderia pseudomallei* (Bps). The bacterium, Bps, has highly recalcitrant, environmental lifestyle and with rapid climatic changes, understanding its dual lifestyle is need of the hour. The samples are from an hyperendemic endemic collected over period of four years that is affected by typhoons, monsoon directions, and other climatic alterations through river flow and canal systems. Through deep genomics study of more than 1000 samples, the authors have identified four dominant lineages and tried to understand their success and transmission through further phylogenetic analysis of dominant lineages, comparative genomic studies involving identifying genes harbouring SNPs and genes unique to the dominant lineages. The authors report the role of horizontal gene transfer in acquisition of new genes for adaptation in extreme environments that is encountered by Bps and recombination in acquisition of new mutations and also genes in the dominant lineage(s). In one of the dominant lineages, the authors also provide evidence through transcriptomic studies that the lineage specific genes harbouring variations or unique genes are important for adaption/fitness rather than virulence/infections indicating environmental persistence in success and transmission route of these dominant lineage(s). Accordingly, the authors also correlate the dominance and movement of lineages according to climatic conditions prevailing in the diverse geographic regions of the country. Overall, the study is a major and novel effort in understanding a fatal pathogen in general and Bps in particular. However below are the major comments that author need to address before any decision is taken.

We extend our gratitude to reviewer #3 for their thoughtful consideration, time, and attention to this paper. The constructive comments provided have significantly enhanced the manuscript. Thank you immensely for your input. All comments have been integrated into the revised manuscript, as marked in the "manuscript with marked changes" file.

1) The coverage of genomes generated in the study is just below 100X. Also, authors need to provide genomic completeness/contamination report as generated by software's like CheckM.

We thank the reviewer for pointing this out. We opted for sequencing at below 100X coverage to strike a balance between cost and study scale. We totally agree with the reviewer regarding the necessity of providing data on genomic completeness and contamination. As a result, we have included the CheckM results in both the Methods (Line 390-392) and Supplementary data 2 to address this aspect.

2) NCBI Genbank accession numbers with annotation of the genome is missing. Just SRA data will not be helpful for the researchers who are not good at bioinformatics and who work on microbiology, genetics, and physiology of the organism.

We totally agree with the reviewer that making the annotated genomes available would be beneficial for the research community. We now deposited the annotated genome through ENA which will be automatically updated on NCBI Genbank. The requested data with the full links listed in Supplementary data 1.

3) Have they confirmed the all the isolates are *B. pseudomallei* by ANI cutoff with the type strain?

We have additionally confirmed that all the studied isolates are *B. pseudomallei* through ANI cut-off, as detailed in the Method section (Line 390-392) and Supplementary data 2.

4) The authors need to work on the title and rest of the manuscript as the findings related to genes unique or under variation in the dominant lineages. Sometime it sounds that the authors are reporting genes unique or under variation in the each or every of the dominant lineages or between two dominant lineages. Hence authors need to clarify the same. For example, Distinct gene content and SNPs described only to 1, 2 lineages ...no inform on such results with other lineages...

We thank the reviewer for this feedback. We have extensively revised the manuscript to ensure a clearer presentation of whether the genes are unique to a single or shared across multiple dominant lineages.

5) While the phylogenomic evidence for dominant lineages is clear, the findings related to unique genes and lineage specific genes carrying SNPs is from one particular dominant lineage, while transcriptomic studies are from another dominant lineage and mapping to a genomic island from another dominant lineages. While it is important to understand the evolution and transmission as revealed by dominant lineages, the authors must not end up confusing. Hence there is need to make the results/discussion clear.

We are grateful for the reviewer's feedback which helped us identify the source of confusion. To address this issue, we have revised the text to ensure that the gene name in K96243 serves as a representative gene of its lineage (lineage 1), while the homologue of K96243 is now appropriately referred to in the discussion related to other dominant linages (lineage 2 and 3) (Line 283-285). We hope these adjustments have enhanced the overall readability of the manuscript.

6) Throughout the sampling period, the four dominant lineages consistently dominated in terms of size and persistence, indicating their robust and stable presence in the sampled population (Fig. 1d)." Authors need to provide year wise phylogenetic trees (may be as supplementary) to indicate the presence of these dominant lineages as the Fig. 1d is not clear.

We thank the reviewer for this comment and have included the year-wise phylogeny of each dominant lineage in the new Supplementary Figures 3 to 5. These figures specifically present the year-wise phylogeny of dominant lineages 1, 2, 3 and their sub-lineages, respectively.

7) Also, the authors can clarify the year wise transmission pattern of dominant lineages to see if their hypothesis based on all the isolates across year holds true.

We thank the reviewer for this comment and apologise for the absence of a time-calibrated phylogeny for dominant lineages in our previous version.

Due to the relatively short sampling period in our study, we were unable to discern the temporal signals for all dominant lineages except for sub-lineage 1.3 (see Supplementary Figure 3). Consequently, we generated a time-calibrated phylogeny specifically for this sub-lineage. Our new analysis indicated that the most recent common ancestor of sub-lineage 1.3 emerged around 2011 (95% HPD of 2000 - 2014). This temporal framework enables us to associate the observed transmission patterns of the sub-lineage with factors such as the frequency of regular monsoons or regional floods, which likely influenced the bacterial transmission dynamics.

Given that the most recent common ancestors of isolates within the parental dominant lineage predate those within its sub-lineage, we anticipate that the transmission history of the dominant lineage was shaped by a higher frequency of events, such as the frequency regular northeast monsoon. This hypothesis is potentially applicable to all isolates. We have extensively revised these new study findings and addressed their limitations (Line 171 - 180).

8) Similarly, recombination studies, mapping of the unique genes/lineage specific genes in dominant lineages in each year is something missing in the study.

We apologise for this oversight and acknowledge that the previous version lacked the association of lineage-specific genes and recombination patterns. We have rectified this by providing a more comprehensive investigation of lineage-specific genes and recombination patterns for each dominant lineage, as detailed in Table 1 and Supplementary Figure 8.

9) Also their mention and generation of complete genome sequence (again in one or two dominant lineage) is not clear and incomplete. Also authors need to provide details regarding obtaining complete genome sequence using nanopore data.

We greatly appreciate the feedback concerning the preparation of the complete reference genome for each lineage. In response, we have incorporated this information in the revised methods section (line 445-462).

10) Overall, the authors have planned an important and innovative study in a lethal pathogen, however there is scope to add on with further more analysis possible with their data. With the kind of year wise resource and data, the authors have not completely justified with analysis and sounds at many places in-complete/abrupt.

We are incredibly grateful to the reviewer's comments. We have explored the possibility of generating the time-calibrated phylogeny at the sub-lineage level. This allowed us to establish a historical timeline between the most recent common ancestor and our sampling period. This timeline helps us correlate events such as monsoons and floods in the region, thereby reinforcing the connection between climate patterns and the dissemination of *B. pseudomallei*. We have also made significant revisions to enhance the clarity and the flow of the text. Once more, we extend our heartfelt gratitude to the reviewer for their valuable time and consideration.

REVIEWER COMMENTS

Reviewer #3 (Remarks to the Author):

The authors have addressed the concerns in the revised manuscript.

Reviewer #4 (Remarks to the Author):

Major points:

Reviewer #1

The authors have sufficiently addressed the comments from reviewer #1. The reviewer's assessment was not very critical of the paper, and the comments appeared easy to address by the authors. I think that the critique from reviewer #1 has helped improve the manuscript.

Reviewer #2

Overall, I think the authors have mostly addressed the comments by reviewer #2. I agree with Reviewer 2's initial assessment that the transcriptomic data does not mesh well with the genomic data. I think the additional transcriptomic data helps the authors address this concern but I think they've somewhat over-inflated the significance of their transcriptional analysis. As it currently stands, the conclusions regarding the selective expression of lineage specific genes is a little weak. I would encourage the authors to include a little more discussion of their transcriptome results in comparison to previous literature (e.g. <https://www.nature.com/articles/s41467-021-22169-1>).

Minor points:

Reviewer #1 comment 5. K96243 is the prototypic genome for *B. pseudomallei* and using this as the reference is the standard for population genetics studies in this bacterium. The authors are correct in using this genome as the reference.

Reviewer #1 comment 8. The addition of additional transcriptomic data is commendable and provides at least a little (much needed) functional investigation in this manuscript.

Reviewer #2: Re The main issue is that one cannot make firm conclusions about transmission of an environmentally mediated pathogen based on clinical isolates alone since human-human transmission is extremely rare.

I agree with the authors here. Precisely because there is no human-to-human transmission, clinical cases can be used to assess environmental distribution, especially if careful case histories are taken.

Reviewer #3 (Remarks to the Author):

The authors have addressed the concerns in the revised manuscript.

We appreciate the reviewer's feedback and are delighted to learn that reviewer agrees that all concerns have been addressed in the revised manuscript.

Reviewer #4 (Remarks to the Author):

Major points:

Reviewer #1

The authors have sufficiently addressed the comments from reviewer #1. The reviewer's assessment was not very critical of the paper, and the comments appeared easy to address by the authors. I think that the critique from reviewer #1 has helped improve the manuscript.

We thank the reviewer for this comment.

Reviewer #2

Overall, I think the authors have mostly addressed the comments by reviewer #2. I agree with Reviewer 2's initial assessment that the transcriptomic data does not mesh well with the genomic data. I think the additional transcriptomic data helps the authors address this concern but I think they've somewhat over-inflated the significance of their transcriptional analysis. As it currently stands, the conclusions regarding the selective expression of lineage specific genes is a little weak. I would encourage the authors to include a little more discussion of their transcriptome results in comparison to previous literature (e.g. <https://www.nature.com/articles/s41467-021-22169-1>).

We appreciate the reviewer's suggestion and agree that due to the limited number of lineage-specific genes carried by the representative isolates, as well as the restricted number of conditions tested in the transcription analysis, we can only elucidate a small proportion of the mechanisms underlying the success of each lineage, rather than the complete picture. We now present this transcription work as a potential avenue for further data exploration, rather than providing a firm conclusion regarding the success of each lineage. We have appropriately toned down this aspect (Line 224-234)

Additionally, we have now highlighted the work of Heacock-Kang et al. The *Burkholderia pseudomallei* intracellular 'TRANSITome' *Nature Communications* in 2021. This study serves as a prototype for transcription experiments that capture different stages of infections, allowing for the comprehensive assessment of genes required during infections. We acknowledged the limitation of using a single condition to represent the environmental condition and propose to use a spectrum of conditions portraying a range of environments in our next study (Line 318-321)

Minor points:

Reviewer #1 comment 5. K96243 is the prototypic genome for *B. pseudomallei* and using this as the reference is the standard for population genetics studies in this bacterium. The authors are correct in using this genome as the reference.

Thank you to the reviewer for validating the use of K96243 as the reference genome of *B. pseudomallei*

Reviewer #1 comment 8. The addition of additional transcriptomic data is commendable and provides at least a little (much needed) functional investigation in this manuscript.

We appreciate the reviewer's positive responses to our attempt to integrate transcriptomic data into our study. However, we acknowledge the limitation of the number of lineage-specific genes carried by the representative strains, which provided some functional investigation but not entirely comprehensive. This caveat has been addressed in Lines 216-217, and 310-321.

Reviewer #2: The main issue is that one cannot make firm conclusions about transmission of an environmentally mediated pathogen based on clinical isolates alone since human-human transmission is extremely rare. I agree with the authors here. Precisely because there is no human-to-human transmission, clinical cases can be used to assess environmental distribution, especially if careful case histories are taken.

We thank reviewer #4 for supporting our rationale on the use of clinical samples to outline the dissemination of *B. pseudomallei* across different geographies. As infection resulted from direct environmental exposure, it is important to note that all clinical isolates were once environmental samples. Based on the available environmental isolates, we further demonstrated that both environmental and clinical isolates can be identified within the same major lineages (Supplementary Figure S2). This finding confirms the validity of using clinical isolates as a proxy to trace bacterial transmission (Line 290-301).

Moreover, we have moderated our message to account for the possibility of transmission patterns being influenced by multiple factors. This adjustment allows for a more open interpretation of the observed patterns, which will be reinforced by additional supporting evidence in future studies (Lines 154-155 and 172-176).